# An efficient targeted nuclease strategy for high-resolution mapping of DNA binding sites

Peter J Skene, Steven Henikoff*

Howard Hughes Medical Institute, Basic Sciences Division, Fred Hutchinson Cancer Research Center, Seattle, United States

**Abstract** We describe Cleavage Under Targets and Release Using Nuclease (CUT&RUN), a chromatin profiling strategy in which antibody-targeted controlled cleavage by micrococcal nuclease releases specific protein-DNA complexes into the supernatant for paired-end DNA sequencing. Unlike Chromatin Immunoprecipitation (ChIP), which fragments and solubilizes total chromatin, CUT&RUN is performed in situ, allowing for both quantitative high-resolution chromatin mapping and probing of the local chromatin environment. When applied to yeast and human nuclei, CUT&RUN yielded precise transcription factor profiles while avoiding crosslinking and solubilization issues. CUT&RUN is simple to perform and is inherently robust, with extremely low backgrounds requiring only ~1/10th the sequencing depth as ChIP, making CUT&RUN especially cost-effective for transcription factor and chromatin profiling. When used in conjunction with native ChIP-seq and applied to human CTCF, CUT&RUN mapped directional long range contact sites at high resolution. We conclude that in situ mapping of protein-DNA interactions by CUT&RUN is an attractive alternative to ChIP-seq.

*For correspondence: steveh@fhcrc.org

**Competing interests:** The authors declare that no competing interests exist.

## Introduction

The action of transcription factors (TFs) at their binding sites on DNA drives gene expression patterns, and so genome-wide TF mapping has become a central goal both for individual researchers and large-scale infrastructural projects. TF profiling is most commonly carried out using chromatin immunoprecipitation (ChIP), a protocol that has changed little since it was first introduced over 30 years ago (*Solomon and Varshavsky, 1985*). Cells are crosslinked with formaldehyde, chromatin is fragmented and solubilized, an antibody is added, and the antibody-bound chromatin is recovered for DNA extraction. Successive advances in DNA mapping technologies have revolutionized the use of X-ChIP (formaldehyde crosslinking ChIP), and with ChIP-seq, base-pair resolution mapping of TFs became feasible (*Rhee and Pugh, 2011*; *Skene and Henikoff, 2015*; *He et al., 2015*). Improvements to ChIP-seq retain the crosslinking step to preserve the in vivo pattern while the entire genome is fragmented to create a soluble extract for immunoprecipitation. However, crosslinking can promote epitope masking and can generate false positive binding sites (*Teytelman et al., 2013*; *Park et al., 2013*; *Jain et al., 2015*; *Baranello et al., 2016*; *Meyer and Liu, 2014*).

ChIP can also be performed without crosslinking, using ionic conditions that do not disrupt electrostatic contacts (*Kasinathan et al., 2014*). 'Native' ChIP provides a map of direct protein-DNA interactions with sensitivity and specificity trade-offs that compare favorably with X-ChIP methods. Native ChIP also minimizes problems with epitope masking and improves efficiency relative to X-ChIP, making it more amenable to low starting numbers of cells (*O'Neill et al., 2006*; *Brind'Amour et al., 2015*). But problems remain with incomplete extraction efficiency of protein-DNA complexes and the potential loss of binding.

**eLife digest** The DNA in a person's skin cell will contain the same genes as the DNA in their muscle or brain cells. However, these cells have different identities because different genes are active in skin, muscle and brain cells. Proteins called transcription factors dictate the patterns of gene activation in the different kinds of cells by binding to DNA and switching nearby genes on or off. Transcription factors interact with other proteins such as histones that help to package DNA into a structure known as chromatin. Together, transcription factors, histones and other chromatin-associated proteins determine whether or not nearby genes are active.

Sometimes transcription factors and other chromatin-associated proteins bind to the wrong sites on DNA; this situation can lead to diseases in humans, such as cancer. This is one of the many reasons why researchers are interested in working out where specific DNA-binding proteins are located in different situations. A technique called chromatin immunoprecipitation (or ChIP for short) can be used to achieve this goal, yet despite being one of the most widely used techniques in molecular biology, ChIP is hampered by numerous problems. As such, many researchers are keen to find alternative approaches.

Skene and Henikoff have now developed a new method, called CUT&RUN (which is short for "Cleavage Under Targets & Release Using Nuclease") to map specific interactions between protein and DNA in a way that overcomes some of the problems with ChIP. In CUT&RUN, unlike in ChIP, the DNA in the starting cells does not need to be broken up first; this means that protein-DNA interactions are more likely to be maintained in their natural state. With CUT&RUN, as in ChIP, a specific antibody identifies the protein of interest. But in CUT&RUN, this antibody binds to the target protein in intact cells and cuts out the DNA that the protein is bound to, releasing the DNA fragment from the cell. This new strategy allows the DNA fragments to be sequenced and identified more efficiently than is currently possible with ChIP.

Skene and Henikoff showed that their new method could more accurately identify where transcription factors bind to DNA from yeast and human cells. CUT&RUN also identified a specific histone that is rarely found in yeast chromatin and the technique can be used with a small number of starting cells. Given the advantages that CUT&RUN offers over ChIP, Skene and Henikoff anticipate that the method will be viewed as a cost-effective and versatile alternative to ChIP. In future, the method could be automated so that multiple analyses can be performed at once.

The uncertainties caused by systematic biases and artifacts in ChIP emphasize the need for methods based on different principles. An important class of non-ChIP mapping methods involves tethering of an enzyme to a DNA-binding protein by a chimeric fusion and action of the enzyme on DNA in the local vicinity. For example, in DamID (*van Steensel et al., 2001*) and related methods (*Southall et al., 2013*; *Hass et al., 2015*), *Escherichia coli* Dam methyltransferase is tethered to the TF and catalyzes $N^6$-methylation of adenine at GATC sites in vivo. Sites can be mapped genome-wide using an $N^6$-methyl-directed restriction enzyme. However, as the resolution of DamID is limited by the distribution of GATC sites, DamID cannot obtain the high resolution that is potentially attainable using a sequencing read-out (*Aughey and Southall, 2016*). An alternative enzyme tethering method, chromatin endogenous cleavage (ChEC) tethers the endo-exonuclease Micrococcal Nuclease (MNase) to the TF (*Schmid et al., 2004*). In ChEC, MNase is activated by permeabilizing cells and adding calcium for controlled cleavage. We recently applied an Illumina sequencing read-out to ChEC (ChEC-seq), achieving near base-pair resolution (*Zentner et al., 2015*). Enzyme tethering methods fundamentally differ from ChIP because they are carried out in vivo (DamID) or in situ (ChEC), with extraction of DNA directly from live or permeabilized cells, thus eliminating the need to solubilize and recover chromatin.

Both DamID and ChEC require that a different chimeric fusion construct be produced for each TF to be mapped, limiting their transferability, for example to animal models, patient biopsies and post-translational modifications. In the original chromatin immunocleavage (ChIC) method, crude nuclei from crosslinked cells are first treated with a TF-specific antibody, followed by addition of a chimeric fusion between Protein A and MNase (pA-MN) and activation by calcium (*Schmid et al.,*

*2004*). Protein A binds specifically to Immunoglobulin G, which obviates the need for a fusion protein. Here we report a major development of ChIC that retains the advantages of enzyme tethering methods, while extending its applicability and ease-of-use to a point that it replaces other existing methodologies. A key feature of our protocol is that in the absence of crosslinking, seconds after calcium-induced MNase cleavage on both sides of the TF, the TF-DNA complex is released into solution, allowing for recovery of pure TF-bound DNA fragments for sequencing simply by centrifugation and DNA extraction. By carrying out the procedure on magnetic beads, our 'Cleavage Under Targets and Release Using Nuclease' (CUT&RUN) technique is simpler than ChIP-seq while retaining the advantages of in situ methods. Targeted digestion by CUT&RUN greatly reduces background relative to complete genomic fragmentation for ChIP, requiring only ~1/10th the sequencing depth of standard ChIP methods. Simple spike-in controls allow accurate quantification of protein binding not possible by other methods. Furthermore, the method allows low starting cell numbers, and robotic automation is possible by performing the reaction on magnetic beads.

## Results

### Overview of the strategy

Chromatin Immuno-Cleavage (ChIC) has the advantage of using TF-specific antibodies to tether MNase and cleave only at binding sites. To adapt ChIC for deep sequencing, it was necessary to reduce the representation of background breaks in DNA that otherwise dominate deep sequencing libraries. The CUT&RUN modifications of ChIC were motivated by the observation that light MNase treatment of nuclei liberates mononucleosomes and TF-DNA complexes, leaving behind oligonucleosomes (*Sanders, 1978*; *Teves and Henikoff, 2012*). We expected that targeted cleavage on both sides of a TF would release the TF-DNA complex into the supernatant, leaving the remainder of the genome in the pelleted nuclei. By performing brief digestion reactions on ice, we would recover TF-DNA complexes in the supernatant before TF-bound MNase diffuses around the genome and cleaves accessible chromatin.

Based on this rationale, we developed a simple CUT&RUN protocol (*Figure 1A*). Unfixed nuclei are (1) immobilized on lectin-coated magnetic beads, (2) successively incubated with antibodies and protein A-MNase (pA-MN) followed by minimal washing steps, (3) mixed with Ca$^{++}$ on ice to initiate the cleavage reaction then stopped seconds-to-minutes later by chelation, and (4) centrifuged to recover the supernatant containing the released TF-DNA complexes. DNA is then extracted from the supernatant and used directly for sequencing library preparation.

### CUT&RUN produces limit digestion of chromatin complexes

We first performed CUT&RUN using crude yeast nuclei. To rigorously compare CUT&RUN and ChIP-seq, we used the same FLAG-tagged TF strains, the same nuclear preparation protocol, the same mouse anti-FLAG monoclonal antibody and the same procedure for Illumina library preparation and paired-end sequencing (*Kasinathan et al., 2014*). As mouse Protein A binds only weakly to mouse IgG, we used a rabbit anti-mouse secondary antibody for CUT&RUN.

To test the efficiency of CUT&RUN, we used a *Saccharomyces cerevisiae* strain expressing 3XFLAG-tagged histone H2A, which would be expected to release nucleosomal fragments genome-wide. Indeed, over a 100-fold digestion time course at 0°C, we observed gradual cleavage and release of fragments down to mononucleosomal size, completely dependent on the presence of the primary antibody (*Figure 1B*).

We next applied CUT&RUN to two structurally distinct *S. cerevisiae* TFs, ARS binding factor 1 (Abf1), and rDNA enhancer binding protein 1 (Reb1), obtaining ~2–3 million mapped paired-end reads per sample. We found that the size distributions of mapped fragments were virtually superimposable below ~150 bp for time points between 4 s and 128 s (*Figure 1C*). This close concordance between time points over a 32-fold range suggests that limit digestion of TF-bound fragments occurs rapidly upon addition of Ca$^{++}$, and demonstrates that digestion time is not a critical parameter.

Mapped TF fragment sizes peaked at ~100 bp, in contrast to H2A fragments, which peaked at ~150 bp. As we expect TF complexes to be smaller than ~100 bp, and nucleosomes are ~150 bp, we mapped ≤120 bp and ≥150 bp fragments separately. Time-point profiles show crisp CUT&RUN

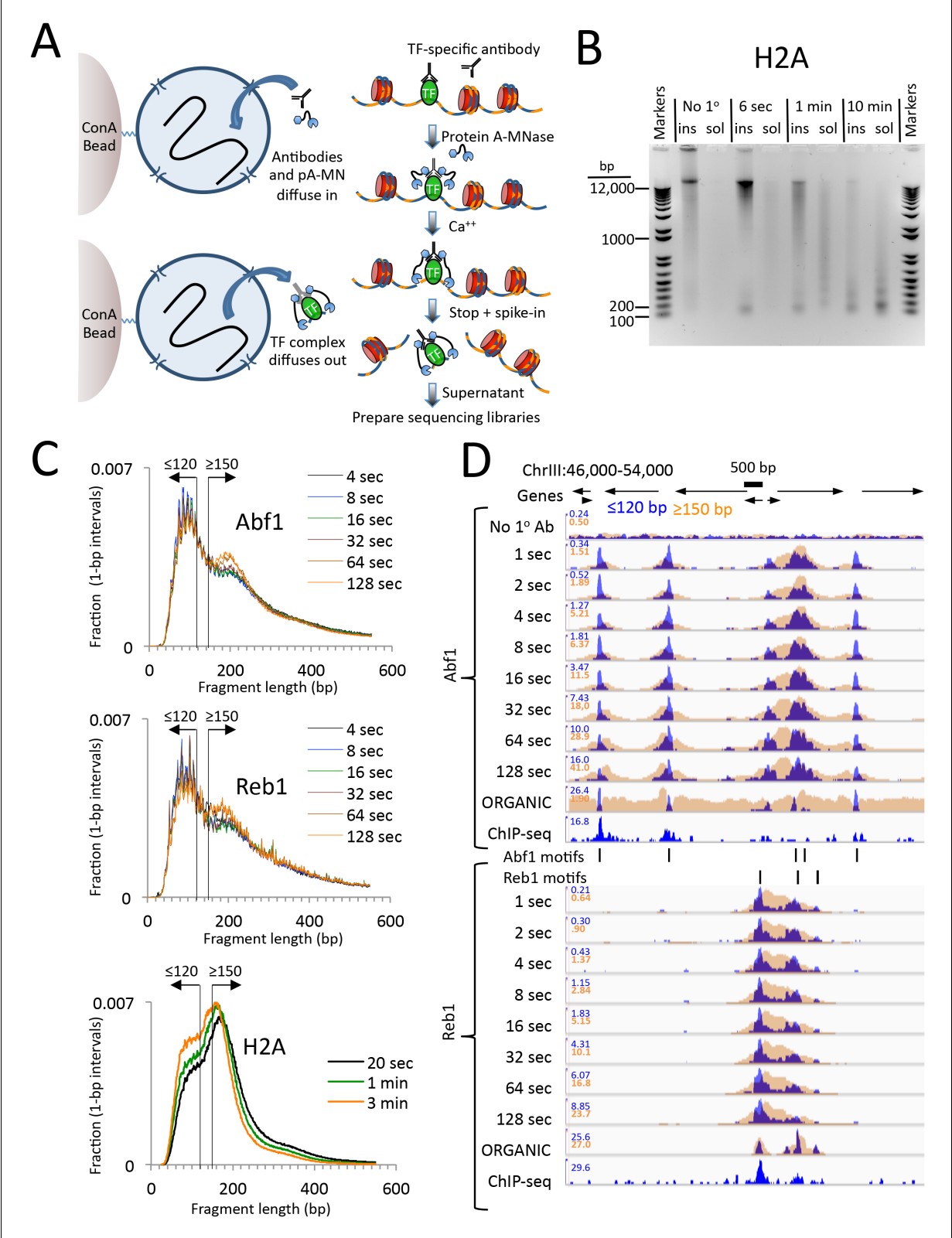

**Figure 1.** CUT&RUN produces limit digestion TF-DNA complexes. (**A**) Schematic diagram of the CUT&RUN strategy. Nuclei attached to magnetic beads can be treated successively with an antibody (or optionally with a primary and secondary antibody) and Protein A-MNase (pA-MN), which diffuse in through the nuclear pores. After Ca$^{++}$ addition to activate MNase cleavage, fragments are released and diffuse out of the nucleus. DNA extracted from the supernatant is used to prepare libraries for paired-end sequencing. (**B**) CUT&RUN cleaves and releases chromatin particles into the

*Figure 1 continued on next page*

*Figure 1 continued*

supernatant. *S. cerevisiae* nuclei in which the endogenous H2A genes were replaced with H2A-3XFLAG were subjected to CUT&RUN and incubated at 0°C in $Ca^{++}$ for the indicated times. DNA extracted from both the insoluble (ins) and soluble (sol) fractions was electrophoresed on a 1% agarose gel. The No 1° Ab control was digested for 10 min in parallel but without having added the primary mouse anti-FLAG antibody. (**C**) Size distributions of mapped paired-end reads from sequencing of indicated TF samples. An H2A size distribution is included for comparison. Data are normalized such that the sum of all points at each length step in base pairs equals 1. (**D**) Time-course profiles for Abf1 and Reb1 samples (~2–3 million mapped paired-end reads per track) showing ≤120 bp (blue) and ≥150 bp (brown) fragment length classes, compared to ORGANIC ChIP-seq (~20–30 million mapped paired-end reads) and standard ChIP-seq (*Paul et al., 2015*) (~5 million Abf1 and ~126 million Reb1 mapped single-end 50 bp reads). A negative control track shows the result of leaving out the primary antibody (No 1° Ab). Within each TF and fragment size group, the Y-axis scale is autoscaled by IGV showing normalized counts and the fragment size classes are superimposed. Ticks mark the location of significant Abf1 (upper) and Reb1 (lower) motifs. This region was chosen as having the largest cluster of Abf1 motifs on Chromosome 3.

The following figure supplements are available for figure 1:

**Figure supplement 1.** CUT&RUN and ORGANIC ChIP produce qualitatively similar TF occupancy profiles.

**Figure supplement 2.** Kinetics of CUT&RUN DNA release.

**Figure supplement 3.** Quantitative recovery of bound TFs in supernatants.

**Figure supplement 4.** Abf1 and Reb1 motifs based on CUT&RUN and ORGANIC ChIP-seq are similar.

peaks within the ≤120 bp size class for each TF motif in each region (*Figure 1D* and *Figure 1—figure supplement 1*). Except for a slow monotonic increase in peak occupancy when normalized to the spike-in control (*Figure 1—figure supplement 2*), no consistent differences between time points were observed within the 1 s to 128 s interval, confirming that gradual release of TF-DNA complexes yields limit digestion reactions. Total DNA extraction and purification of small fragments produced nearly identical results (*Figure 1—figure supplement 3*), which demonstrates that extraction of DNA from the supernatant quantitatively recovers TF-bound fragments.

## CUT&RUN robustly maps yeast TF binding sites in situ at high resolution

To verify that the ≤120 bp fragments represent cleavages around TF binding sites, we first identified all significant Abf1 and Reb1 motifs in the genome and found that motifs based on CUT&RUN data and motifs based on ORGANIC data were nearly identical (*Figure 1—figure supplement 4A–D*). We use ORGANIC-derived motifs to scan the yeast genome, which provided us with a comprehensive list of 1899 Abf1 and 1413 Reb1 motifs determined completely independently of CUT&RUN. We confirmed that the majority of peak calls overlapped the motif for each dataset, with somewhat better performance for CUT&RUN than ORGANIC for Abf1 and vice versa for Reb1 (*Figure 1—figure supplement 4E*). We then aligned the ≤120 bp and ≥150 bp profiles centered over these motifs and constructed heat maps. When rank-ordered by occupancy over the 2 kb interval centered over each Abf1 and Reb1 motif, we observed that >90% of the TF sites were occupied by fragments over the corresponding motif relative to flanking regions (*Figure 2* and *Figure 2—figure supplement 1*, upper panels), representing likely true positives. CUT&RUN occupancies over Abf1 and Reb1 motifs showed high dynamic range relative to nuclease accessibility (*Figure 2—figure supplement 1*, lower panels), seen in heat maps as higher contrast above background for CUT&RUN. In contrast, Abf1 fragments showed negligible occupancy at non-overlapping Reb1 sites and vice-versa for Reb1 fragments at non-overlapping Abf1 sites (*Figure 2* and *Figure 2—figure supplement 1*, middle panels). The almost complete correspondence between the presence of a TF motif and occupancy of the TF, and the general absence at sites of a different TF, imply that CUT&RUN is both highly sensitive and specific for TF binding.

To directly compare CUT&RUN to high-resolution ChIP-seq, we similarly lined up 'ORGANIC' ChIP-seq data over Abf1 and Reb1 motifs. As previously reported (*Kasinathan et al., 2014*), ORGANIC ChIP-seq detected the large majority of Abf1 true positive motifs and nearly all Reb1 motifs throughout the genome (*Figure 2*, upper middle panels). The best Reb1 data were obtained

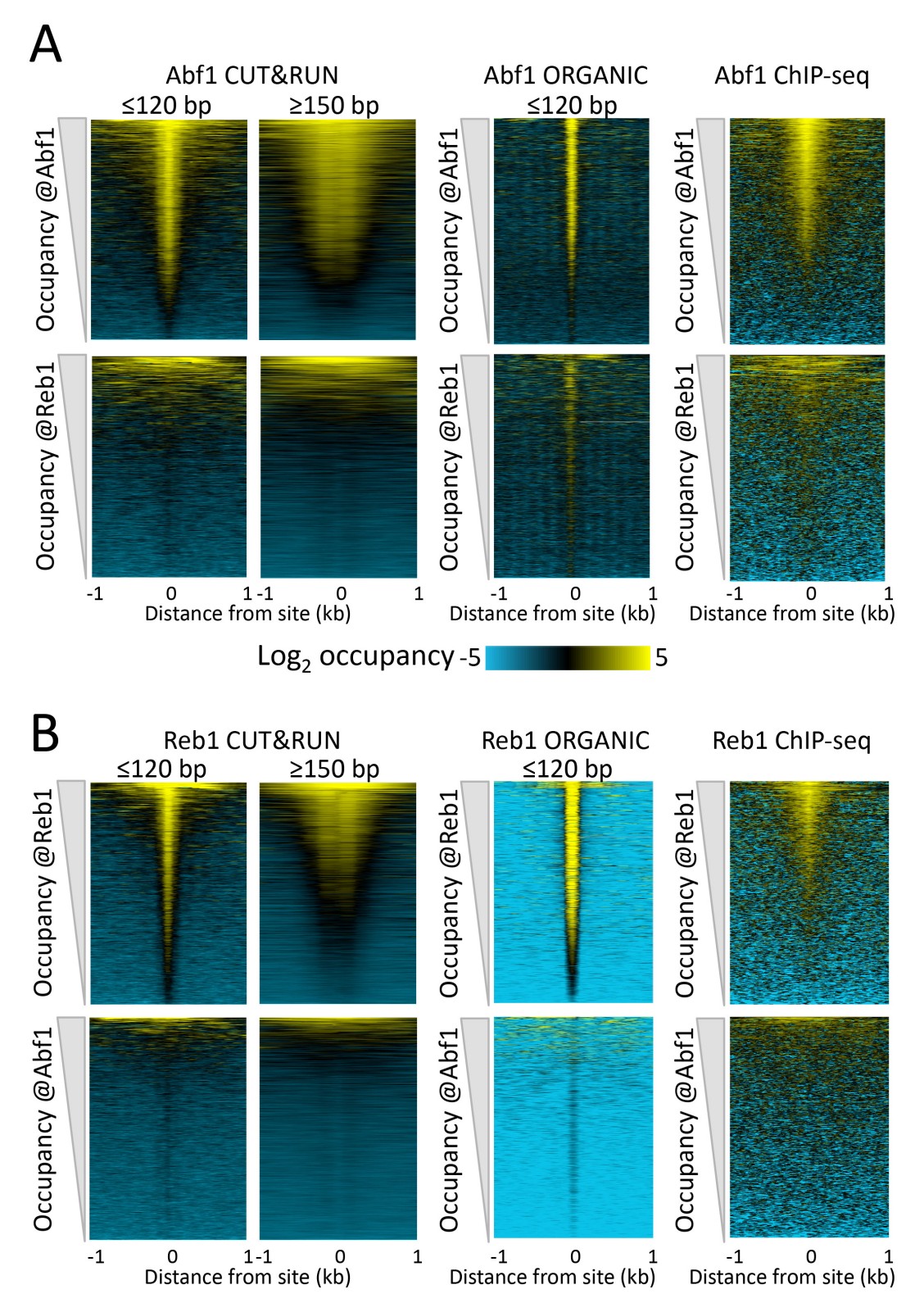

**Figure 2.** CUT&RUN accuracy and robustness compares favorably with ChIP-seq. Abf1 (**A**) and Reb1 (**B**) heat maps of CUT&RUN datasets from a single experiment (20160630), pooling 1" to 32" time-course samples, and separating into ≤120 bp and ≥150 bp size classes (left). Also shown is the ORGANIC ChIP-seq ≤120 bp size class (middle) and standard ChIP-seq datasets (right). Abf1 has two DNA-binding domains spaced ~10 bp apart (*Cho et al., 1995*), whereas Reb1 has a single Myb-like DNA-binding domain (*Morrow et al., 1990*). Solubilization of Abf1 chromatin after MNase

*Figure 2 continued on next page*

Skene and Henikoff. eLife 2017;6:e21856. DOI: 10.7554/eLife.21856

*Figure 2 continued*

digestion required 600 mM NaCl to obtain the best trade-off between specificity and sensitivity, whereas for Reb1, 80 mM gave the best results (*Kasinathan et al., 2014*), and these are the datasets used for comparison. As in our previous comparison of ORGANIC to ChIP-exo and ChIP-chip (*Kasinathan et al., 2014*), we consider the set of all statistically significant Abf1 and Reb1 motifs as the 'gold standard' for judging sensitivity (occupancy of sites by the correct TF) and specificity (exclusion from sites of an incorrect TF). Aligned profiling data were centered and oriented over the motif for the same TF (top) and for the other TF (bottom) for display (removing 81 sites where Abf1 and Reb1 sites were within 50 bp of one another) and were ordered by average pixel density over the −1 kb to +1 kb span of the ≤120 bp datasets using Java Treeview with log$_2$ scaling and contrast = 5. Ordering was performed independently for CUT&RUN (based on ≤120 bp fragments) and ChIP-seq, in which case the approximate fraction of sites occupied relative to flanking regions becomes evident, and comparison of the top panel (correct TF) to the bottom panel (incorrect TF) reflects the sensitivity/specificity tradeoff for a dataset. Sites were determined by MAST searching of the *S. cerevisiae* genome using the position-specific scoring matrices (PSSMs) based on ChIP-seq data (*Figure 1—figure supplement 4*), but similar results were obtained using MAST with PSSMs based on CUT&RUN data (not shown).

The following figure supplement is available for figure 2:

**Figure supplement 1.** CUT&RUN reveals cleavage kinetics in situ.

with 80 mM NaCl extraction, and the best Abf1 data were obtained with 600 mM NaCl, although the dynamic range for Reb1 was always better than that for Abf1 with frequent false positive occupancy (*Figure 2*, lower middle panels). In contrast, CUT&RUN showed the same dynamic range for both TFs over the same range of digestion time points with ~10 fold fewer paired-end reads, demonstrating that CUT&RUN is more robust than ORGANIC ChIP-seq. Relative to these high-resolution methods (*Kasinathan et al., 2014*), standard ChIP-seq using crosslinking and sonication showed inferior sensitivity and specificity (*Figure 2*, right panels). Thus, CUT&RUN provides robust TF occupancy maps with improved sensitivity/specificity trade-offs relative to ChIP-seq.

To estimate the resolution of CUT&RUN, we plotted the 'footprint' of each TF as the average density of fragment ends around the motif midpoint. For both Abf1 and Reb1, we observed sharp 20 bp wide footprints, indicating that these transcription factors protect ~20 bp centered over the motif with near base-pair resolution (*Figure 3A*). Interestingly, upstream and downstream 'slopes' in the cleavage maps show a sawtooth pattern on either side of both Abf1 and Reb1 motifs, with distances between 'teeth' ~10 bp apart over >100 bp, and confirmed by autocorrelation analysis to be independent of base composition (*Figure 3B*). Such 10 bp periodic cleavage preferences match the 10 bp/turn periodicity of B-form DNA, which suggests that the DNA on either side of these bound TFs is spatially oriented such that tethered MNase has preferential access to one face of the DNA double helix. Tethering of MNase to a TF constrains it to cleave nearby DNA even on the surface of a nucleosome, suggesting flexibility of the chromatin fiber (*Figure 3C*). Thus, the very rapid kinetics that we observe at 0°C are due to immobilized MNase poised for cleavage near the tethering site.

## CUT&RUN precisely maps chromatin-associated complexes

High-resolution mapping of mobile components of the chromatin landscape can be challenging for ChIP-based methods. For example, the ~1 megadalton 17-subunit RSC nucleosome remodeling complex dynamically slides a nucleosome that it transiently engulfs (*Lorch et al., 2010*; *Ramachandran et al., 2015*), and the Mot1 DNA translocase dynamically removes TATA-binding protein (TBP) from sites of high-affinity binding (*Zentner and Henikoff, 2013*; *Auble et al., 1997*). Although X-ChIP crosslinks nucleosome remodeling complexes to their nearest nucleosomes, native ChIP successfully captures yeast chromatin remodelers at their sites of action, both in nucleosome-depleted regions (NDRs) and on nucleosomes (*Zentner et al., 2013*). For CUT&RUN to profile such large chromatin-associated complexes we found it necessary to extract total DNA rather than chromatin solubilized by CUT&RUN in situ, which may be too large to diffuse through nuclear pores. Therefore, we extracted all DNA and preferentially removed large DNA fragments with AMPure beads. When this modified protocol was applied to Mot1 over a >2 order-of-magnitude digestion range, we observed chromatin profiles that were very similar to those obtained using ORGANIC profiling, but with only ~15% the number of paired-end reads (*Figure 4A*). As expected, Mot1 peaks on the upstream side of TBP binding sites are seen for both CUT&RUN and ORGANIC profiles, confirming that Mot1 approaches TBP from the upstream side in vivo (*Zentner and Henikoff, 2013*) as it

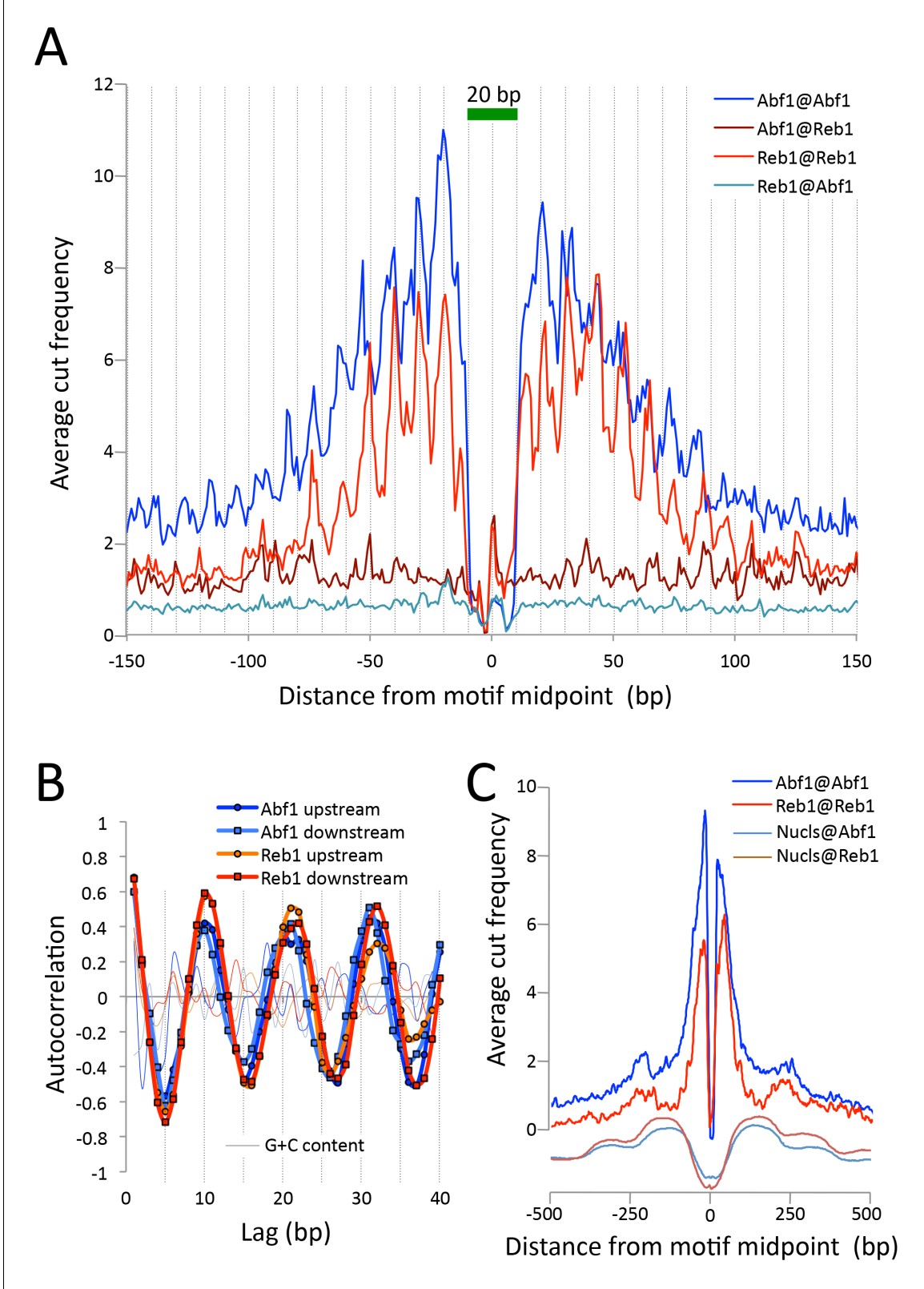

**Figure 3.** CUT&RUN maps TF binding sites at high resolution. (**A**) Mapping of fragment ends reveals a deep 'hole' and steep 'walls' for Abf1 and Reb1 CUT&RUN datasets averaged at their oriented and aligned motifs genome-wide, plotting all normalized base-pair counts from combined 1"−32" datasets (**Figure 2**). Sawtooth patterns with an apparent ~10 bp periodicity on the upstream and downstream 'slopes' are confirmed by (**B**) autocorrelation analysis of the difference between the 1 bp resolution profile shown in (**A**) and the same profile smoothed with an 11 bp sliding

*Figure 3 continued on next page*

*Figure 3 continued*

window, which also shows that there is no corresponding periodicity in average G+C content (thin lines). (C) Same as (A), but subject to smoothing with an 11 bp sliding window and displayed at larger scale. The fact that the slopes around Reb1 show depressions at +150 and −150 likely reflects the presence of phased nucleosomes, shown below (Nucls, Y-axis arbitrary) based on the ≥150 bp size class from ORGANIC input data (*Kasinathan et al., 2014*).

does in vitro (*Wollmann et al., 2011*). Heat map and average plot analyses show that the ≤120 bp fragments track closely with TBP sites, whereas the ≥150 bp fragments are diffusely distributed in the local vicinity, perhaps representing Mot1 translocation dynamics (*Figure 4—figure supplement 1*).

We also applied CUT&RUN to Sth1, the catalytic component of the RSC complex. RSC acts to slide nucleosomes at NDRs, and so we aligned yeast genes at the inferred dyad axis of the +1 nucleosome just downstream of the transcription start site (*Ramachandran et al., 2015*). We observed uniform digestion over a 5 s to 30 min time course (*Figure 4—figure supplement 2A*), and confirmation of the abundance of RSC directly over the GAL4 UAS (*Figure 4—figure supplement 2B*) (*Floer et al., 2010*). Sth1 peaks are most abundant in the NDRs, where CUT&RUN profiles show a gradual increase in yield with digestion times between 5 s and 10 min (*Figure 4B*), indicating that quantitative limit digestions are obtained using our protocol. Importantly, we observed a nearly flat line for our negative control derived from 3XFLAG-Sth1 nuclei treated in parallel for the maximum digestion time, but where the primary anti-FLAG antibody was omitted. Our results for Sth1 CUT&RUN are similar to results for Sth1 ORGANIC profiling (*Ramachandran et al., 2015*), but with much higher yield (*Figure 4C*). We conclude that CUT&RUN provides efficient high-resolution mapping of chromatin-associated complexes, even those that are very large and dynamic.

## CUT&RUN resolves rare insoluble DNA-binding protein complexes

Abf1 and Reb1 are relatively abundant TFs, but many DNA-binding proteins of interest are rare, and so can be challenging to profile by ChIP. In budding yeast, there is only only one centromeric nucleosome per chromosome, which is only ~1% the molar abundance of Abf1 or Reb1. An additional challenge to studying the centromeric nucleosome, which contains the CenH3 (Cse4) histone variant in place of H3, is that it is part of the multi-megadalton kinetochore complex throughout the cell cycle (*Akiyoshi et al., 2010*), which renders it highly insoluble (*Krassovsky et al., 2012*). To profile the Cse4 nucleosome by CUT&RUN, we split the samples after digestion, extracting just the supernatant from one aliquot, and total DNA from the other. In this way we could compare the recovery of the soluble and the insoluble kinetochore complex. In parallel, we similarly profiled histone H2A. By taking the difference between total and soluble chromatin, we can infer the occupancy of each histone in the insoluble pellet. As expected for the insoluble kinetochore, the highest Cse4 occupancy on the chromosome is seen at the centromere (*Figure 5A*). Strikingly, occupancy of insoluble H2A, which is present in every nucleosome througout the genome, is also maximum at the centromere. Indeed, at all 16 yeast centromeres we observe very similar enrichments of Cse4 and H2A confined to the ~120 bp functional centromere over the digestion time-course, with resolution that is 4-fold better than that of standard X-ChIP (*Figure 5B*). We also extracted total DNA from the bead-bound chromatin derived from cells that had been formaldehyde crosslinked prior to applying CUT&RUN, with similar results (*Figure 5C*). Interestingly, crosslinking results in a more distinct profile and the appearance of phased nucleosomes on either side, which we interpret as a reduction in chromatin flexiblity with crosslinking, while demonstrating that the basic strategy can be applied to crosslinked cells.

To confirm that the differences we observed between the CUT&RUN supernatant and total DNA were due to differential solubility of kinetochore chromatin, we split the samples before digestion, and for one aliquot we stopped the cleavage reaction with 2 M NaCl and recovered the supernatant for sequencing. We obtained similar results for the high-salt fraction as for total DNA (*Figure 5—figure supplement 1*). The unequivocal presence of insoluble H2A in the centromeric nucleosome directly addresses the continuing controversy over its composition (*Wisniewski et al., 2014*; *Henikoff et al., 2014*; *Aravamudhan et al., 2013*; *Shivaraju et al., 2012*). Moreover, as the yeast centromeric nucleosome wraps DNA that is >90% A+T (*Krassovsky et al., 2012*), the intactness of

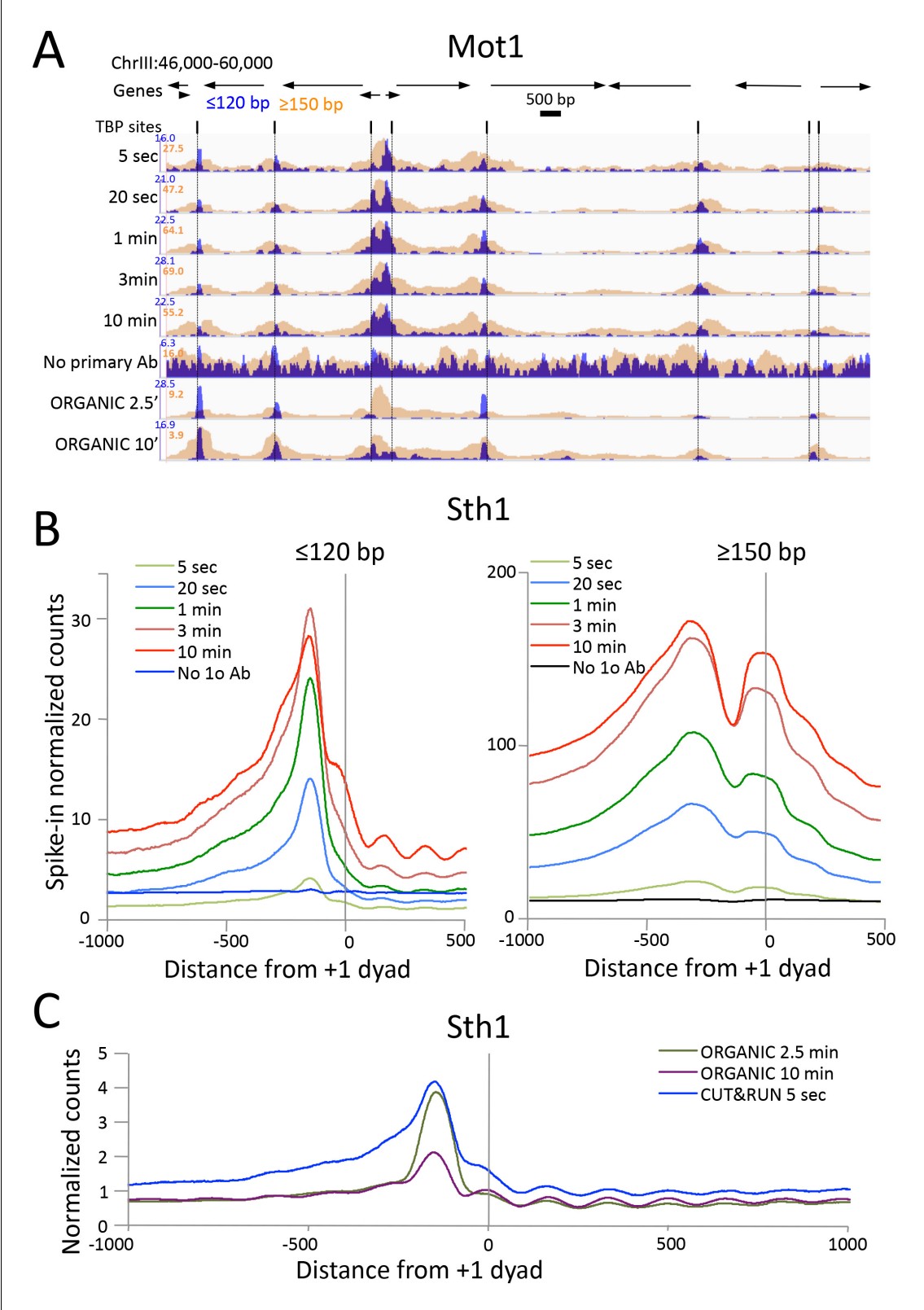

**Figure 4.** CUT&RUN precisely maps large mobile chromatin complexes. (**A**) Representative tracks showing a Mot1 CUT&RUN time-course experiment (average ~3 million paired-end reads per sample), including a no primary antibody (No 1o Ab) negative control, aligned with Mot1 ORGANIC data for two MNase digestion time points (2.5' and 10', average 22 million reads per sample) (*Zentner and Henikoff, 2013*). TBP sites shown as dotted lines reveal that Mot1 peaks are just upstream of TBP peak maxima. (**B**) Occupancy profiles for Sth1 CUT&RUN digestion over a 120-fold range, spike-in

*Figure 4 continued on next page*

*Figure 4 continued*

normalized, showing absolute quantitation. (**C**) Sth1 ORGANIC profiles (~15 million reads) show concordance with the CUT&RUN 5 s sample (~2 million reads). Note that the same CUT&RUN 5 s ≤120 bp profile is shown in both panels (**B**) and (**C**), but at different scales.

The following figure supplements are available for figure 4:

**Figure supplement 1.** CUT&RUN and ORGANIC profiles for Mot1.

**Figure supplement 2.** CUT&RUN and ORGANIC profiles for Sth1.

the centromeric particle over a >100 fold digestion time-course (*Figure 5*) demonstrates that CUT&RUN is not biased by the inherent preference of MNase for AT-rich DNA (*Chung et al., 2010*; *McGhee and Felsenfeld, 1983*). We conclude that CUT&RUN can map large DNA-binding complexes, even those that are rare, insoluble and AT-rich.

## CUT&RUN probes nearby chromatin

Examination of the ≥150 bp profiles (*Figure 1D* and *Figure 4*) reveals broad peaks around the binding sites, sometimes with 'notches' corresponding to the sites themselves that deepen with time of digestion (*Figure 2—figure supplement 1*). We interpret this pattern as representing the gradual release of fragments with one end resulting from cleavage around the TF-DNA complex and a second cleavage that is close enough to the TF-bound site to produce a soluble fragment. Heat map analysis of the ≥150 bp fragments also showed occupancy of Abf1 and Reb1 fragments over their respective binding motifs, extending ~0.5 kb on either side (*Figure 2* and *Figure 2—figure supplement 1*). Such extended local cleavage is reminiscent of the >1 kb reach of DamID (*van Steensel et al., 2001*), which suggests that the flexibility of the tether results in probing of nearby chromatin.

## CUT&RUN maps human transcription factor binding sites at high resolution

Having established proof-of-principle in a simple well-studied genome, we next applied CUT&RUN to CCCTC-binding factor (CTCF) in human K562 cells. To directly compare the efficiency of various methods, we randomly selected 10 million reads for each technique and plotted the raw scores as an indication of information content per sequenced read. As was the case for yeast TFs, CTCF CUT&RUN showed higher dynamic range than other profiling methods, including standard X-ChIP-seq and ChIP-exo (*Figure 6A*). When aligned to CTCF motifs found within DNaseI hypersensitive sites or previously identified binding sites, CUT&RUN and X-ChIP-seq CTCF heat maps showed strong concordance, with CUT&RUN having a higher dynamic range (*Figure 6B*). A no antibody control showed undetectable background (*Figure 6—figure supplement 1*) when CUT&RUN is performed at low temperature (*Figure 6—figure supplement 2*). As was the case for budding yeast TFs, we observed release of the neighboring fragments, which correspond to phased nucleosomes immediately adjacent to CTCF sites. By plotting just the end positions of the short CUT&RUN fragments that are the cleavage positions of the tethered MNase, we observed pronounced 'tram-tracks' separated by 44 bp at defined positions relative to the CTCF motif. Furthermore, the exact cleavage pattern is consistent over a ~300 fold time-course digestion range, with a predominant single base-pair cut site on either side of the CTCF-bound site, highlighting the limit digest obtained (*Figure 6C*). This pattern indicates that the cleavage positions are precise and highly homogeneous within the population of cells. Our results suggest that CUT&RUN accurately maps both the TFs and their flanking chromatin in the same experiment.

CTCF has 11 zinc fingers and therefore may represent an unusually stable protein–DNA interaction. We therefore tested CUT&RUN using Myc and Max which are basic-loop-helix proteins that bind to a short E-box motif and have b residence times (*Phair et al., 2004*). CUT&RUN successfully mapped both Myc and Max at high resolution (*Figure 6—figure supplement 3A*). In the case of Max, a quantitative comparison with ENCODE ChIP-seq data is possible as the same antibody was used, and here CUT&RUN had a much higher dynamic range and therefore was able to robustly identify a much larger number of Max binding sites (*Figure 6—figure supplement 3B*). To bind

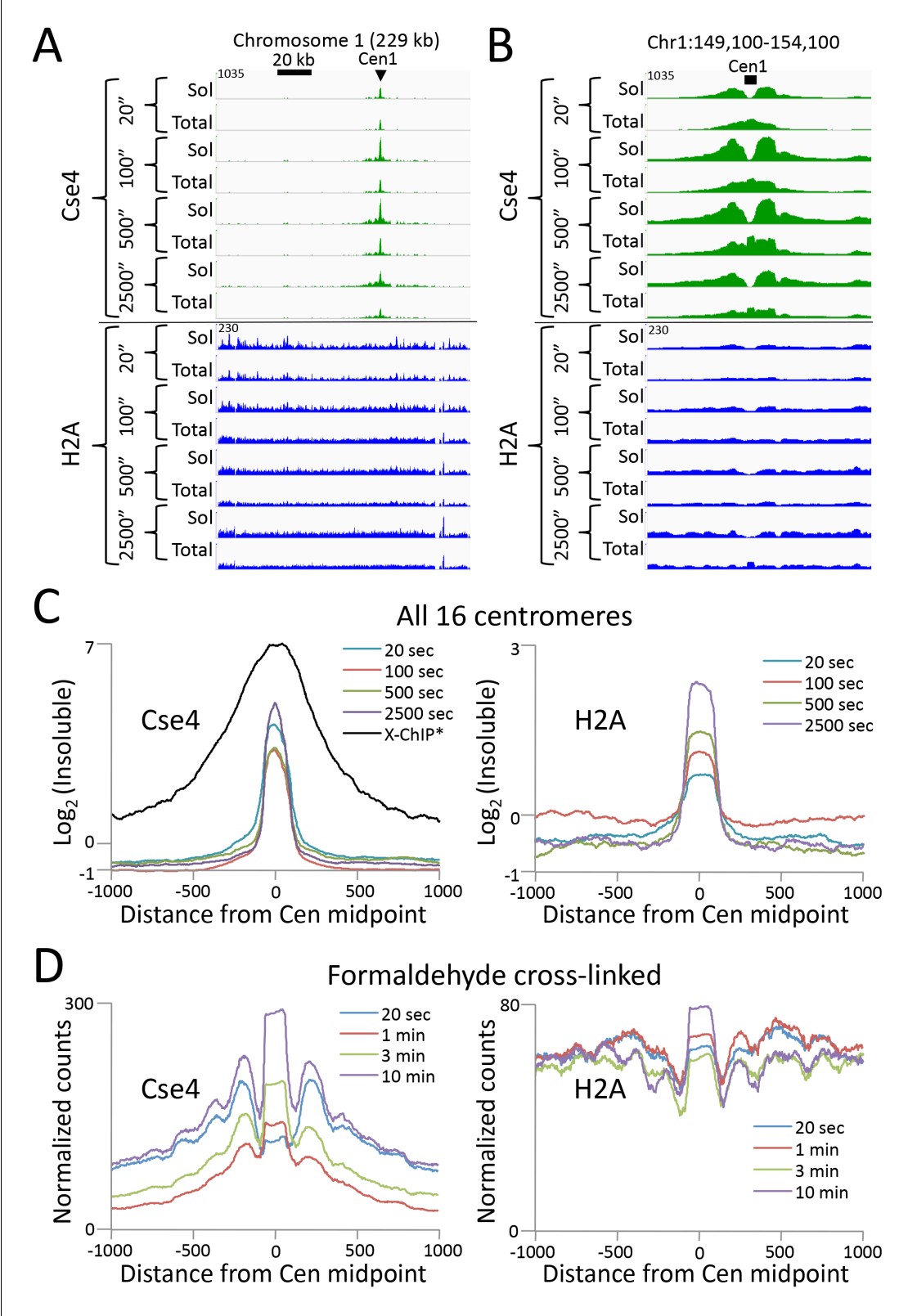

**Figure 5.** CUT&RUN maps the rare highly insoluble *S. cerevisiae* kinetochore complex. (**A**) After stopping digestion for the indicated times, samples were split in half and both the soluble fraction and total DNA were extracted. Large fragments were removed from total DNA with AMPure beads before library preparation. Normalized counts are shown for *S. cerevisiae* Centromere 1, where Cse4 and H2A tracks are on the same Y-axis scale. Similar maxima over centromeres was also seen genome-wide. (**B**) Same as (**A**) but zoomed in over the 5 kb interval at the centromere. (**C**) Occupancies

*Figure 5 continued on next page*

*Figure 5 continued*

of insoluble Cse4 and H2A, where we define log$_2$(Insoluble) = log$_2$(Total) – log$_2$(Soluble) = log$_2$(Total/Soluble) for the medians of all 16 *S. cerevisiae* centromeres aligned around their midpoints. A published X-ChIP-seq profile (*Pekgöz Altunkaya et al., 2016*) is shown on the same scale for comparison (left). Asterisk: log$_2$(ChIP/Input) averaged over two replicates. (D) Normalized count profile of Cse4 and H2A CUT&RUN applied to formaldehyde cross-linked cells digested for the indicated times.

The following figure supplement is available for figure 5:

**Figure supplement 1.** CUT&RUN maps the rare highly insoluble *S. cerevisiae* kinetochore complex.

DNA at E-boxes, Myc forms a heterodimer with Max (*Blackwood et al., 1991*) but in addition Max has other binding partners (*Ayer and Eisenman, 1993*), As expected, we see very high overlap with Max present at almost all Myc binding sites. In contrast, there is poor overlap between previously identified binding sites by ENCODE X-ChIP-seq for Myc and Max, as 10-fold fewer Max sites were identified. However, when we lined up Max ENCODE X-ChIP-seq data over Max CUT&RUN sites, we saw high occupancy (*Figure 6—figure supplement 3C*), suggesting that the lower dynamic range of X-ChIP-seq relative to CUT&RUN was responsible for the failure to identify these Max binding sites by X-ChIP-seq.

## CUT&RUN maps histone modifications in compacted chromatin

We also considered the possibility that antibody-tethered MNase may be excluded from highly compacted heterochromatic regions in higher eukaryotes and as such CUT&RUN might be limited to analysis of protein-DNA interactions in euchromatic regions. We therefore performed CUT&RUN for the repressive histone mark H3K27me3. By analyzing 10 million reads from CUT&RUN and ENCODE X-ChIP-seq, we observed similar H3K27me3 landscapes, but at a much higher dynamic range for CUT&RUN, which demonstrates that Protein A-MNase is able to access compacted chromatin (*Figure 6—figure supplement 4*). Furthermore, H3K27me3 cleaved chromatin is readily released from the intact nuclei into the soluble fraction, indicating that CUT&RUN is applicable to probing protein-DNA interactions in compacted chromatin.

## CUT&RUN directionally maps long-range genomic contacts

As nucleosome-sized fragments adjacent to TFs are released together with TF-containing fragments, we wondered whether 3D adjacencies might also be subject to cleavage and release. Chromosome-Conformation-Capture (3C) methods, such as Hi-C and ChIA-PET (*Tang et al., 2015*; *Lieberman-Aiden et al., 2009*), are the preferred techniques for mapping 3D genome-wide contacts. These methods use the same formaldehyde crosslinking protocol as X-ChIP to identify 3D interactions, such as between a TF bound at an enhancer and its contact with a promoter via co-activators. In this example the binding sites for a protein identified by X-ChIP will include both the promoter and the enhancer, even though one of the interactions is via indirect protein-protein interactions crosslinked by formaldehyde. But in both X-ChIP and 3C-based mapping there is no systematic way to distinguish between direct and indirect sites. We therefore attempted to map CTCF binding sites using native ChIP, which we have previously shown results in mapping only direct binding sites containing the TF-specific DNA-binding motif, due to the transient nature of protein-protein interactions (*Kasinathan et al., 2014*). We developed a new native ChIP protocol (Appendix 1), and achieved near-complete protein extraction with no evidence of protein redistribution (*Figure 7—figure supplement 1*). Under native conditions, we identified 2298 sites with high motif scores. In contrast, CUT&RUN mapping of CTCF detected ~22,000 sites that were also present in X-ChIP (*Figure 6—figure supplement 1*), with a diverse range of motif scores (*Figure 7—figure supplement 2*). As expected, all sites identified by native ChIP also were robustly detected by CUT&RUN and X-ChIP, showing a similar signal distribution (*Figure 7A*). CUT&RUN sites lacking a significant native ChIP signal nevertheless showed a robust footprint in the native ChIP input with a similar cumulative distribution of counts (*Figure 7B*), indicating the presence of unknown bound factors, as would be expected for 3D genomic interactions. This suggests that CUT&RUN, as with X-ChIP, can discover both direct (native CTCF peak) and indirect (CUT&RUN peak only) chromatin interactions at high resolution.

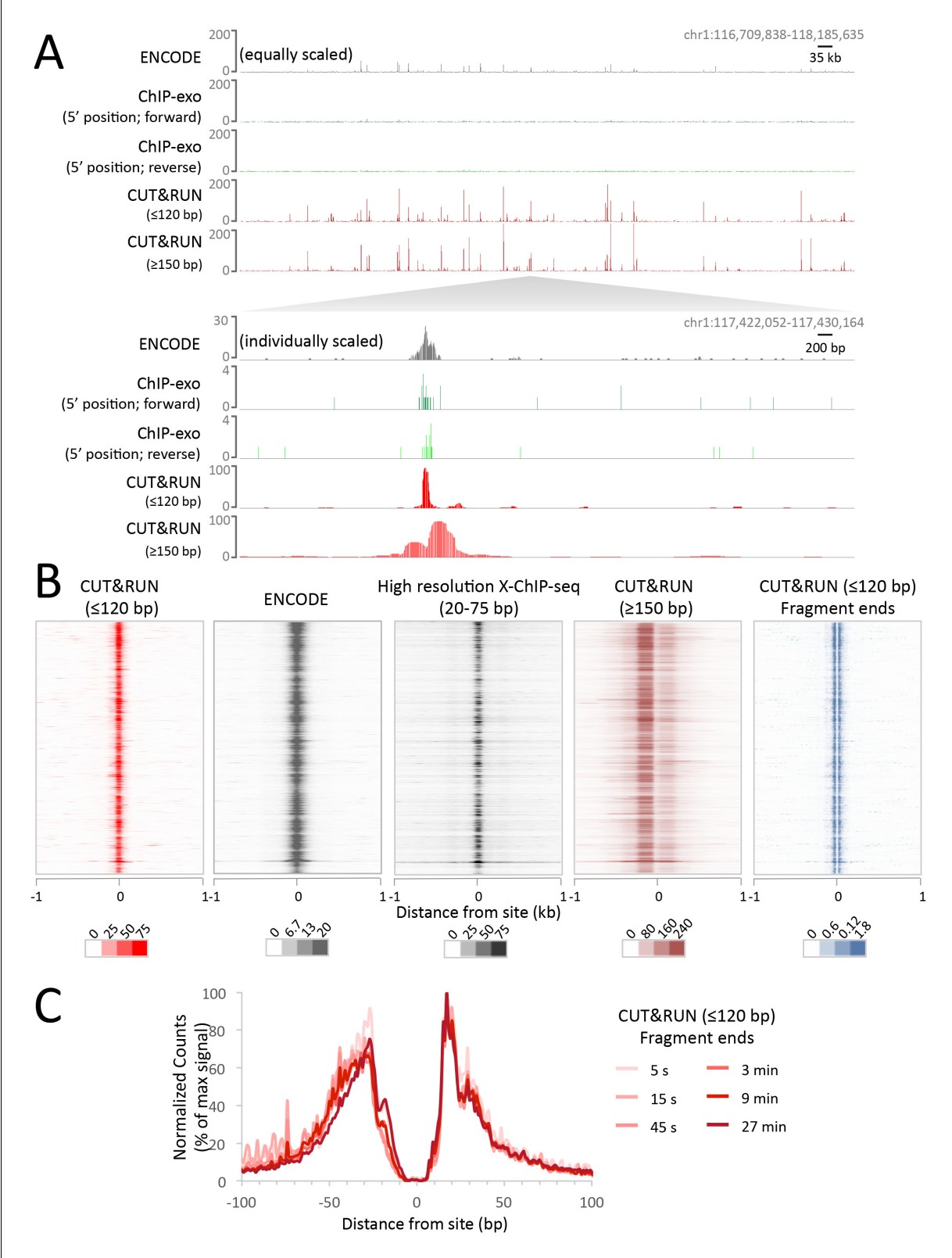

**Figure 6.** CUT&RUN maps high-resolution footprints of CTCF. (**A**) Representative signal over a genomic locus for 10 million randomly sampled reads from ENCODE CTCF ChIP (GSM749690), CTCF ChIP-exo, and CUT&RUN. In the top panel, the y-axis is the same for all datasets indicating the higher dynamic range for CUT&RUN. In the bottom panel, the y-axis is individually set. (**B**) Heat maps of CUT&RUN pooled datasets (7.5 min to 45 min) separated into ≤120 bp (including fragment ends) and ≥150 bp size classes and of ENCODE X-ChIP-seq and high resolution X-ChIP-seq (*Skene and*

*Figure 6 continued on next page*

*Figure 6 continued*

***Henikoff, 2015***) for CTCF in human K562 cells. Sites were determined by an unbiased approach in which the data were centered and oriented on CTCF motifs that were found within DNaseI hypersensitive sites and ordered by genomic location. Asymmetric release of the upstream and downstream nucleosome likely comes from epitope location controlling access to nucleosomes either side of the motif. (**C**) Mean plots of end positions from ≤120 bp fragments resulting from a CUT&RUN digestion time-course centered over sites as above. Data are represented as a percentage of the maximum signal within the ±1 kb flanking region.

The following figure supplements are available for figure 6:

**Figure supplement 1.** CUT&RUN recapitulates X-ChIP-seq but with higher dynamic range.

**Figure supplement 2.** CUT&RUN has low background when performed on ice.

**Figure supplement 3.** The high signal-to-noise ratio of CUT&RUN allows robust identification of DNA binding sites not possible with X-ChIP-seq.

**Figure supplement 4.** CUT&RUN can map compacted chromatin with a high dynamic range.

To confirm that CTCF CUT&RUN sites not observed by native ChIP correspond to contact sites, we compared direct and indirect sites to contact sites observed by ChIA-PET. CTCF ChIA-PET identifies interacting genomic regions mediated through CTCF, but cannot discern between directly CTCF bound regions and the interacting indirectly bound region. We find that for a typical ~1 Mb genomic region *all* high-scoring ChIA-PET fragments overlap with direct and indirect sites (***Figure 8A***). Whereas mapped CTCF ChIA-PET fusion fragments are in the several kb range, determined by the distance between sites for the 6-cutter restriction enzyme used, both direct and indirect CUT&RUN CTCF sites are mapped with near base-pair resolution. Moreover, 91% of the direct sites are present in CTCF ChIA-PET data, with 43% of these ChIA-PET fragments interacting with an indirect site, and the remainder contained a high CUT&RUN signal (***Figure 8C***), which suggests these are indirect sites involved in multiple contacts, just below the peak calling threshold.

As further evidence that CUT&RUN can detect indirect contact interactions, we found a high frequency of Hi-C interactions between direct sites and indirect sites and a quantitative correlation between Hi-C score and CUT&RUN signal at the indirect sites (***Figure 8B***). Therefore, by comparing CUT&RUN and native ChIP it is possible to map contact sites at near base-pair resolution, to distinguish direct from indirect protein binding sites that result from long-range genomic interactions, and to determine the directionality to these contacts, not feasible by other methods.

## CUT&RUN allows quantitative measurements with low cell numbers

Typical ChIP-seq experiments require large numbers of cells, and low cell number ChIP has been limited to abundant proteins (***Kasinathan et al., 2014***; ***Brind'Amour et al., 2015***). We performed CTCF CUT&RUN with starting K562 cell numbers ranging from 600,000 to 10 million. To compare absolute occupancies between datasets, we used a simple spike-in strategy (see Materials and methods), allowing accurate quantitative measurements of protein occupancy. When normalized to spike-in DNA, we observed that the number of cleavage events is proportional to the starting cell number (***Figure 9***). Furthermore, when the data are normalized to the total number of reads aligning to the human genome, there is no clear difference in the samples, suggesting that high data quality is maintained with low input material.

## Discussion

### A simple method for chromatin profiling

CUT&RUN is based on the ChIC antibody-tethered nuclease strategy of Laemmli and co-workers (***Schmid et al., 2004***). To adapt ChIC into a genome-wide profiling method, we made five critical modifications. First, we immobilize permeabilized cells or crude nuclei to magnetic beads, allowing for rapid and efficient solution changes, so that CUT&RUN is performed in a day and is suitable for automation. Second, we bind antibodies and pA-MNase to native unfixed nuclei, where epitopes

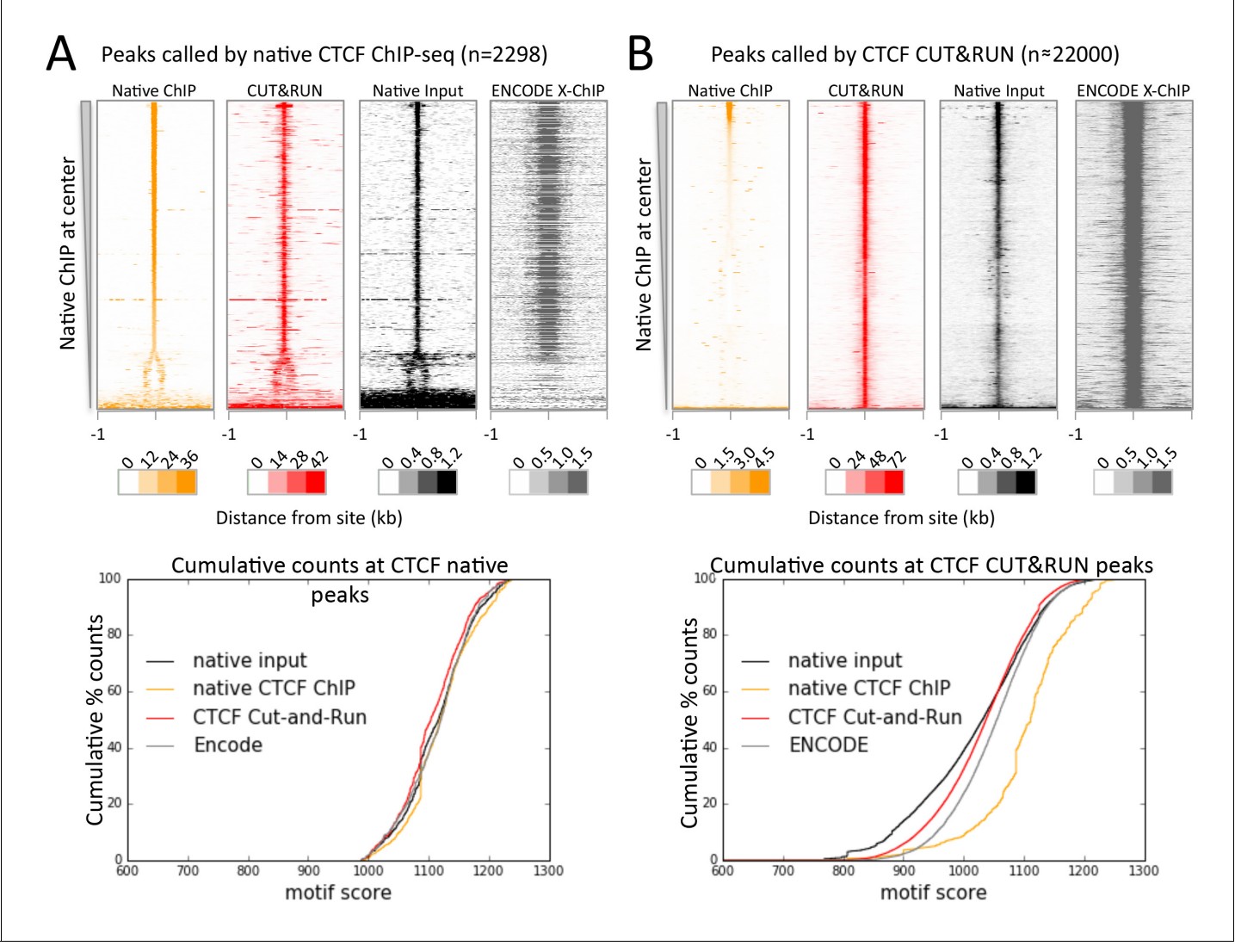

**Figure 7.** CTCF directly binds a subset of CUT&RUN peaks despite a robust footprint at all sites. (**A**) Chromatin was fragmented and solubilized under native conditions and either directly sequenced as native input or CTCF bound chromatin was immunoprecipitated and sequenced. ENCODE X-ChIP-seq was analyzed for comparative purposes. Peaks of CTCF binding under native conditions were identified and centered on the best match to the CTCF motif (JASPAR database MA0139.1, http://jaspar.genereg.net/). Data were plotted over these sites (−1 to +1 kb) as heat maps for native ChIP DNA fragments (20–75 bp) and CUT&RUN (≤120 bp) and ordered by native CTCF ChIP occupancy (sum over the center region (−30 to +30 bp) minus the sum over the flanks (−1000 to −700 and +700 to+1000 bp). The graph below shows the cumulative percent of sequencing counts for the different techniques over peak-called sites (−30 to +30 bp) and ranked by similarity to the CTCF motif. This shows the high concordance between the chromatin profiling techniques at native ChIP peaks. Note that the dynamic range scales for Native ChIP and CUT&RUN are ~30–40 fold higher than those for Native Input and ENCODE X-ChIP, which was needed to show the input and ENCODE patterns. (**B**) Data plotted over CUT&RUN peak-called sites, with processing as per (**A**). The cumulative distribution shows the shift to lower motif scores for CUT&RUN sites (see *Figure 7—figure supplement 2* and the separation between CUT&RUN and native ChIP.

The following figure supplements are available for figure 7:

**Figure supplement 1.** A modified native ChIP protocol allows complete protein extraction.

**Figure supplement 2.** Peaks identified by CUT&RUN have a more diverse range of motif scores than peaks from native ChIP.

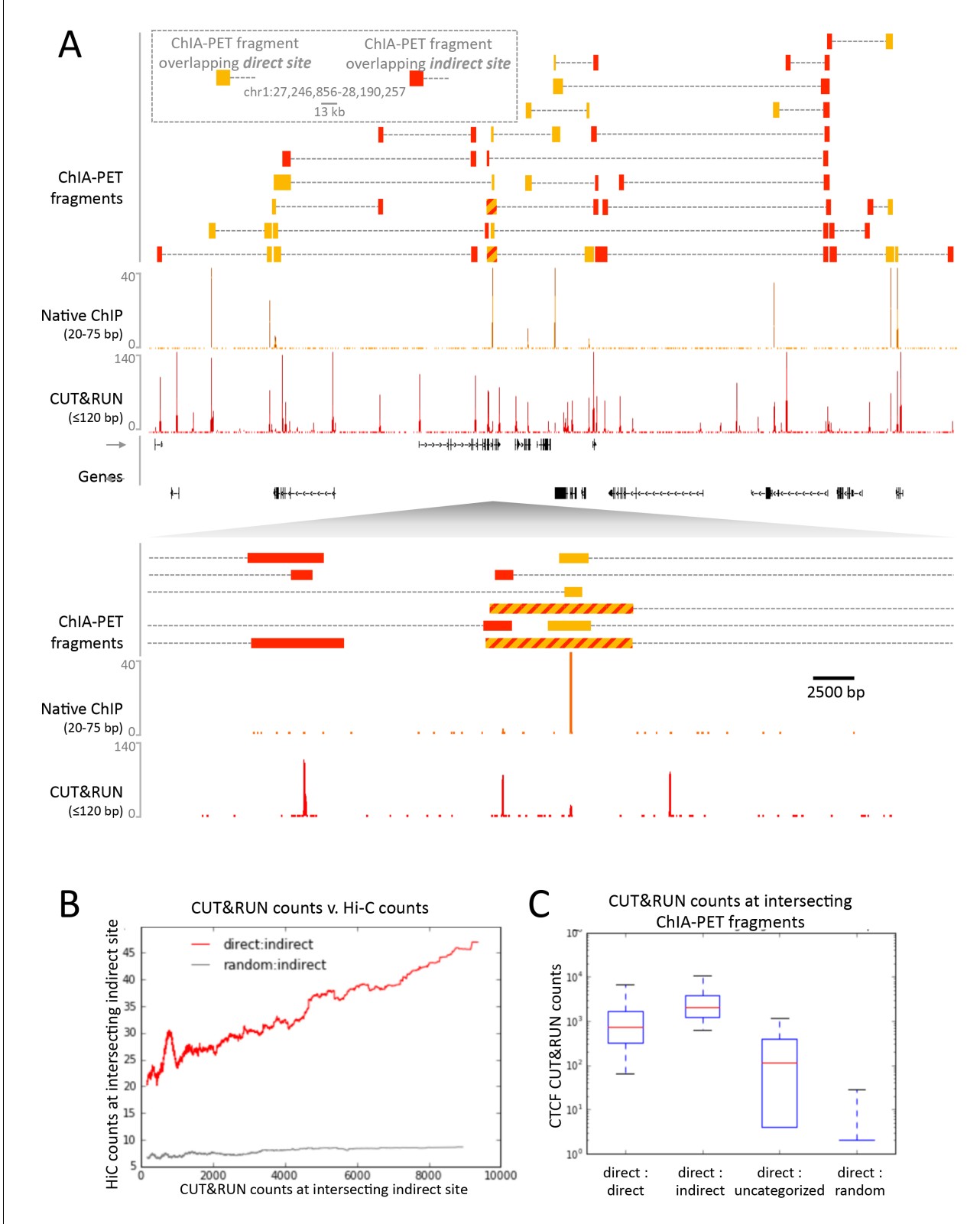

**Figure 8.** CUT&RUN in combination with native ChIP can discern direct and indirect 3D contact sites. (**A**) Typical genomic region displaying CUT&RUN (≤120 bp), native ChIP (20–75 bp) data for CTCF and CTCF ChIA-PET fragments (GSM1872886; score ≥15). ChIA-PET fragments were ascribed as a direct interaction (overlapping a native ChIP peak) or an indirect interaction (overlapping a CUT&RUN peak only). (**B**) Peak called sites were separated into either direct (present in native ChIP) or indirect (only present in CUT&RUN). Hi-C fragments that intersect with direct sites or an equal number of

*Figure 8 continued on next page*

*Figure 8 continued*

random genomic locations were identified. The Hi-C interacting fragment was then intersected with the indirect sites and the CUT&RUN signal compared to Hi-C raw signal. Data were ranked by CUT&RUN score and plotted as a moving average with a window size of 1500. (C) ChIA-PET fragments that contained a direct site were identified and the interacting fragment intersected with direct peaks, indirect peaks or random locations as above. Interacting fragments that did not overlap with these sites were classed as uncategorized. Boxplots indicate the CUT&RUN score for the observed contacts at the interacting fragment.

are preserved and accessible. Third, as cleavage by immobilized MNase is a zero-order reaction, we performed digestion at ice-cold temperature, which limits diffusion of released fragments, thus reducing background. Fourth, we use native chromatin, which allows us to fractionate cleaved fragments based on solubility (*Sanders, 1978*; *Teves and Henikoff, 2012*; *Jahan et al., 2016*) to enrich specifically for the released chromatin complex. We simply remove the insoluble bulk chromatin, and only chromatin fragments with cleavages on both sides of a particle enter the supernatant. Fifth, after DNA extraction we use these soluble fragments for Illumina library preparation and paired-end DNA sequencing. We find that CUT&RUN performs as well as or better than ChIP-seq with respect to simplicity, resolution, robustness, efficiency, data quality, and applicability to highly insoluble complexes. CUT&RUN requires only ~1/10th of the sequencing depth of other high resolution methodologies due to the intrinsically low background achieved by performing the reaction in situ. As nuclei are intact when MNase is activated, CUT&RUN can probe the local environment around the targeted site. Indeed, we find that CUT&RUN recovers sites of 3D contacts with base-pair resolution at relatively low sequencing depth in human cells.

## CUT&RUN is widely applicable

Although ChIC was described as a basic mapping method using Southern blotting 12 years ago, we are unaware of a single publication using it. Meanwhile, ChIP-seq alone has been mentioned in ~30,000 publications for profiling almost every type of chromatin component, including histone modifications, transcription factors and chromatin-associated proteins. Like ChIP, CUT&RUN is antibody-based, so that it can be applied to any epitope on chromatin, making it a general method for chromatin profiling that takes advantage of the large antibody production infrastructure developed for ChIP. CUT&RUN provides quantitative occupancy profiles with standard and spike-in normalization options implemented by our custom software for processing and comparing ChIP-seq and CUT&RUN datasets. The only non-standard feature of CUT&RUN is the requirement for the pA-MN fusion protein, which can be produced and purified in a batch from bacterial culture that yields enough pA-MN for profiling >100,000 samples.

As CUT&RUN is based on different principles from ChIP, it can resolve crosslinking-, sonication- and solubilization-related issues. Backgrounds are low with CUT&RUN, because cleavages occur only around binding sites, whereas ChIP first pulverizes the entire genome, and these fragments contribute to a genome-wide background noise that must still be sequenced. The near absence of detectable background under the brief low-temperature conditions that we used, the lack of preference for accessible or AT-rich DNA, and the recovery of essentially all Abf1 and Reb1 motifs in the yeast genome, suggests that CUT&RUN is not subject to the types of artifacts that sometimes have plagued ChIP (*Teytelman et al., 2013*; *Park et al., 2013*; *Jain et al., 2015*; *Baranello et al., 2016*; *Kasinathan et al., 2014*). Furthermore, CUT&RUN antibody binding takes place in an intact nuclear environment resembling conditions for immunofluorescence microscopy, so that it should be successful for antibodies that are validated cytologically, even those that fail in ChIP. As CUT&RUN solubilizes chromatin only after the targeted cleavage reaction, it should be appropriate for extending classical chromatin salt-fractionation (*Sanders, 1978*; *Teves and Henikoff, 2012*; *Jahan et al., 2016*) to specific TFs and chromatin complexes.

## CUT&RUN precisely maps long-range contacts

A consequence of using intact nuclei for CUT&RUN is that the long reach of antibody-tethered MNase can probe the local environment. In yeast, we observed cleavages on one surface of the DNA flanking the TF, gradually decreasing with distance. In human cells, we observed cleavages at sites previously identified as contact sites for CTCF. Recently, Hi-C contact sites have been predicted

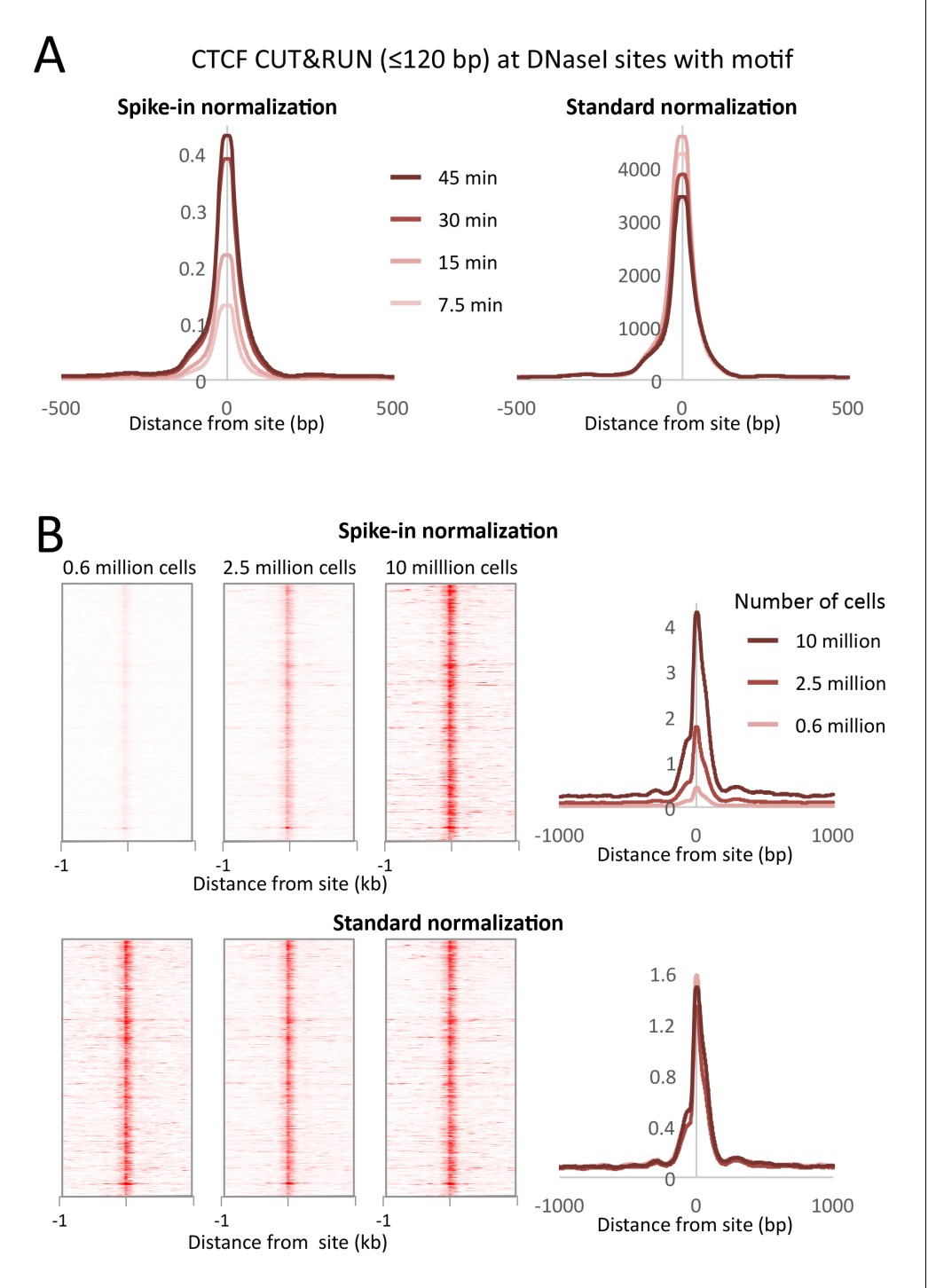

**Figure 9.** CUT&RUN allows simple quantification of protein-DNA interactions. (**A**) A digestion timecourse of CUT&RUN was performed for CTCF in K562 cells. To allow quantification of released fragments, 1 ng of Drosophila DNA was added after the cleavage reaction. Mean plots of ≤120 bp sequenced fragments were centered over CTCF motifs found within DNaseI sites. Data were normalized either to the number of fly reads (Spike-in normalization) or to the total number of human reads (Standard normalization). (**B**) A titration of starting material was used to map CTCF binding genome-wide. Heat maps and mean plots were generated for the ≤120 bp sequenced fragments using Spike-in or Standard normalization. Data were centered over CTCF motifs found within DNaseI sites.

computationally with a high degree of confidence given sites of CTCF binding (*Sanborn et al., 2015*). As CUT&RUN maps both CTCF binding sites and interactions, and our native ChIP protocol identifies those sites that are directly TF-bound, it can provide a complete high-resolution 1D map of a genome while enriching its 3D contact map with high-resolution distinctions between direct and indirect TF binding sites.

### Low background levels reduce sequencing costs

ChIP-seq analysis typically includes normalization to compensate for different numbers of reads between samples. In ChIP-seq, whole genome fragmentation leads to a constant low density genome-wide background that provides a basis for normalization, for example, in comparing wild-type and knockdown cell lines. Although normalization fails on abundant proteins, this can be corrected by the use of spike-in controls (*Bonhoure et al., 2014*; *Chen et al., 2015*; *Orlando et al., 2014*). However, a rigorous spike-in strategy requires the addition of cells from a different species, and quantification is reliant upon antibody cross-reactivity (*Orlando et al., 2014*). To normalize between samples despite the low background in CUT&RUN, we find that simply adding a constant low amount of fragmented spike-in DNA from a different species suffices and allows accurate quantification of protein occupancy.

The low background levels of cleavage with CUT&RUN require fewer reads to crisply define peaks. For example only ~10 million paired-end reads were required for each CTCF time point, similar to the requirement for low-resolution ChIP-seq, and many fewer than for ChIP-exo, which required ~100 million reads for CTCF (*Rhee and Pugh, 2011*). Furthermore, in the cases of Max and H3K27me3, 10 million reads for CUT&RUN provided very high dynamic range, but 10 million reads were insufficient for calling peaks from Max ENCODE X-ChIP-seq. This cost-effectiveness makes CUT&RUN attractive as a replacement for ChIP-seq especially where depth of sequencing is limiting. We might attribute the high efficiency of CUT&RUN to fundamental differences between in situ profiling and ChIP: CUT&RUN retains the in vivo 3D conformation, so antibodies access only exposed surfaces in a first-order binding reaction, whereas in ChIP, antibodies interact with the solubilizable genome-wide content of the pulverized cells or nuclei. Furthermore, CUT&RUN cleavage is effectively a zero-order reaction, resulting in steady particle release during the brief low temperature time-course for all bound epitopes in the genome. Accounting for epitope abundances, we estimate that our mapping of ~22,000 direct and indirect CTCF sites with 600,000 cells is comparable to the sensitivity of ultra-low input ChIP-seq protocols that are typically restricted to abundant histone modifications, such as H3K27me3, with ~5000 cells (*Brind'Amour et al., 2015*). Whereas ultra-low-input ChIP provides only ~2 kb resolution, CUT&RUN provides near base-pair resolution. The inherent robustness, high information content, low input and sequencing requirements and suitability for automation of our method suggests that CUT&RUN profiling of CTCF and other TFs might be applied to epigenome diagnostics.

In summary, CUT&RUN has a number of practical advantages over ChIP and its derivatives: with low background resulting in low sequence depth requirements, the ease of use makes it amenable to robotic automation, while allowing accurate quantification with a simple spike-in strategy. We conclude that in all important respects CUT&RUN provides an attractive alternative to ChIP-based strategies.

## Materials and methods

### Biological

W1588-4C *S. cerevisiae* strains carrying Flag-tagged H2A (SBY2688), Cse4 (SBY5146), Abf1 and Reb1 under the control of their respective endogenous promoters were previously described (*Kasinathan et al., 2014*; *Krassovsky et al., 2012*; *Gelbart et al., 2001*). Yeast nuclei were prepared as described (*Kasinathan et al., 2014*), flash frozen in 0.5–0.6 ml aliquots and stored at −80°C. Human K562 cells were cultured under standard conditions. Standard protocols were used for electrophoretic gel analysis and immunoblotting. Antibodies used were Mouse anti-FLAG (M2, Sigma, St. Louis, MO, Catalog #F1804), Rabbit anti-mouse (Abcam, Cambridge, UK, Catalog #ab46540), CTCF (Millipore Billerica, MA, Catalog #07–729), H3K27me3 (Millipore Catalog #07-449), c-Myc (Cell Signaling Technology Beverly, MA, Catalog #D3N8F), Max (Santa Cruz Biotechnology,

Dallas, TX, Catalog #sc-197) and RNA Pol II (8WG16, Abcam Catalog #ab817). The pK19pA-MN plasmid was a gift from Ulrich Laemmli and pA-MN protein was prepared from *E. coli* cells as described (*Schmid et al., 2004*).

## CUT&RUN for yeast nuclei

CUT&RUN begins with crude nuclei prepared according to published procedures. The following protocol is provided in step-by-step format (Appendix 2). Nuclei from ~5 $\times$ $10^8$ cells at OD600 ~0.7 were prepared as described (*Orsi et al., 2015*), divided into 10 600 μL aliquots, snap-frozen and held at −80°C, then thawed on ice before use. Bio-Mag Plus Concanavalin A (lectin) coated beads were equilibrated with HNT (20 mM HEPES pH7.5, 100 mM NaCl and 0.1% Tween 20) that was supplemented with 1 mM each $MgCl_2$, $CaCl_2$ and $MnCl_2$. Only $Ca^{++}$ and $Mn^{++}$ are needed to activate lectins, and omitting $MgCl_2$ had no effect on binding of permeabilized cells to beads. The beads (300 μL) were rapidly mixed with a thawed nuclei aliquot and held at room temperature (RT) ≥5 min, placed on a magnet stand to clear (<1 min), and decanted on a magnet stand. The beads were then incubated 5 min RT in HNT supplemented with protease inhibitors (Roche Complete tablets) and 1 mM phenylmethylsulfonyl fluoride (PMSF) (=HNT-PPi) containing 3% bovine serum albumen (BSA) and 2 mM EDTA pH 8, then incubated 5′ with HNT-PPi +0.1% BSA (blocking buffer), using the magnet stand to decant. The beads were incubated 2 hr at 4°C with mouse anti-FLAG antibody (1:200–1:350), decanted, washed once in HNT + PMSF, then incubated 1 hr at 4°C with rabbit anti-mouse IgG antibody (1:200) in blocking buffer. The beads were washed once in HNT + PMSF, then incubated 1 hr at 4°C with pA-MN (600 μg/ml, 1:200) in blocking buffer. The beads were washed twice in HNT + PMSF and once in 20 mM HEPES pH 7.5, 100 mM NaCl (Digestion buffer), optionally including 10% polyethylene glycol 8000 for Sth1 and Mot1. The beads were brought up in 1.2 ml Digestion buffer, divided into 8 $\times$ 150 μL aliquots, equilibrated to 0°C, then quickly mixed with $CaCl_2$, stopping the reaction with 150 μL 2XSTOP [200 mM NaCl, 20 mM EDTA, 4 mM EGTA, 50 μg/ml RNase A (Thermo Scientific, Waltham, MA, Catalog #EN0531) and 40 μg/ml glycogen (Sigma, Catalog #10901393001), containing 5-50 pg/ml heterologous mostly mononucleosome-sized DNA fragments extracted from formaldehyde crosslinked MNase-treated Drosophila chromatin as a spike-in]. After incubating at 37°C for 20 min, the beads were centrifuged 5 min at 13,000 rpm at 4°C, the supernatant was removed on a magnet stand and mixed with 3 μL 10% SDS and 2 μL Proteinase K (Invitrogen, Carlsbad, CA, Catalog #25530049), incubated at 70°C 10 min, then extracted at room temperature once with buffered phenol-chloroform-isoamyl alcohol (25:24:1, Sigma P2069), transferred to a phase-lock tube (Qiagen, Hilden, Germany, Catalog #129046), re-extracted with one vol $CHCl_3$, transferred to a fresh tube containing 2 μL 2 mg/ml glycogen, precipitated by addition of 2–2.5 vol ethanol, chilled in ice and centrifuged 10 min at 13,000 rpm at 4°C. The pellet was rinsed with 100% ethanol, air-dried and dissolved in 25 μL 0.1 x TE8 (=1 mM Tris pH 8, 0.1 mM EDTA).

To extend CUT&RUN for high-salt extraction, digestions were performed in a 50 μL volume, stopped with 50 μL 2XSTOP, omitting RNase and substituting the standard 200 mM NaCl with 4 M NaCl. After 20 min at 37°C, 200 μL 67 μg/ml RNase A was added, incubated 20 min, then centrifuged 13,000 rpm to clarify the supernatant.

## CUT&RUN for mammalian cells

Human K562 cells were purchased from ATCC (Manassas, VA, Catalog #CCL-243). CUT&RUN was performed using a centrifugation-based protocol. Ten million cells were harvested by centrifugation (600 g, 3 min in a swinging bucket rotor) and washed in ice cold phosphate-buffered saline (PBS). Nuclei were isolated by hypotonic lysis in 1 ml NE1 (20 mM HEPES-KOH pH 7.9; 10 mM KCl; 1 mM $MgCl_2$; 0.1% Triton X-100; 20% Glycerol) for 5 min on ice followed by centrifugation as above. (We have found that nucleases in some cells cause $Mg^{++}$-dependent degradation of DNA, in which case 0.5 mM spermidine can be substituted for 1 mM $MgCl_2$.) Nuclei were briefly washed in 1.5 ml Buffer 1 (20 mM HEPES pH 7.5; 150 mM NaCl; 2 mM EDTA; 0.5 mM Spermidine; 0.1% BSA) and then washed in 1.5 ml Buffer 2 (20 mM HEPES pH 7.5; 150 mM NaCl; 0.5 mM Spermidine; 0.1% BSA). Nuclei were resuspended in 500 μl Buffer 2 and 10 μl antibody was added and incubated at 4°C for 2 hr. Nuclei were washed 3 x in 1 ml Buffer 2 to remove unbound antibody. Nuclei were resupended in 300 μl Buffer 2 and 5 μl pA-MN added and incubated at 4°C for 1 hr. Nuclei were washed 3 x in 0.5 ml Buffer 2 to remove unbound pA-MN. Tubes were placed in a metal block in ice-water and

quickly mixed with 100 mM $CaCl_2$ to a final concentration of 2 mM. The reaction was quenched by the addition of EDTA and EGTA to a final concentration of 10 mM and 20 mM respectively and 1 ng of mononucleosome-sized DNA fragments from Drosophila DNA added as a spike-in. Cleaved fragments were liberated into the supernatant by incubating the nuclei at 4°C for 1 hr, and nuclei were pelleted by centrifugation as above. DNA fragments were extracted from the supernatant and used for the construction of sequencing libraries. We have also adapted this protocol for use with magnetic beads (Appendix 3).

## Spike-in normalization

Genome-wide background in TF ChIP-seq datasets is typically sufficiently high to provide a constant background level for normalization to compensate for variations between samples in library preparation and sequencing. For standard normalization, the number of fragment ends corresponding to each base position in the genome was divided by the total number of read ends mapped. However, the inherently low background levels of CUT&RUN necessitate a spike-in control for quantitative comparisons (*Hu et al., 2014*). For spike-in normalization of human CUT&RUN, we added a low constant amount of *Drosophila melanogaster* DNA to each reaction. We mapped the paired-end reads to both human and fly genomes, normalizing human profiles to the number of fly reads (*Figure 9*). Using internal normalization, we observed no increase in cleavages over a digestion time-course. However, by normalizing to the fly spike-in DNA, we observed an ~4 fold increase in cleavage level over time. As such, CUT&RUN is amenable to accurate quantification of protein-DNA interactions.

## Library preparation, sequencing and data processing

Sequencing libraries were prepared from DNA fragments as described (*Kasinathan et al., 2014*; *Henikoff et al., 2011*) but without size-selection, following the KAPA DNA polymerase library preparation kit protocol (https://www.kapabiosystems.com/product-applications/products/next-generation-sequencing-2/dna-library-preparation/kapa-hyper-prep-kits/) and amplifying for eight or more cycles. To deplete total DNA samples of large fragments originating from insoluble chromatin, samples were mixed with ½ volume of Agencourt AMPure XP beads, held 5–10 min, placed on a magnet stand, and the supernatant was retained, discarding the beads. To reduce the representation of the remaining large fragments, the number of PCR cycles using the KAPA polymerase library preparation method was increased to 14 cycles and adapter concentrations were increased accordingly: Increasing the number of PCR cycles favors exponential amplification of shorter fragments over linear amplification of fragments that are too long for polymerase to completely transit. Libraries were sequenced for 25 cycles in paired-end mode on the Illumina HiSeq 2500 platform at the Fred Hutchinson Cancer Research Center Genomics Shared Resource. Paired-end fragments were mapped to the sacCer3/V64 genome and build and to release r5.51 (May 2013) of the *D. melanogaster* genomic sequence obtained from FlyBase using Novoalign (Novocraft) as described to generate SAM files. For human samples, paired-end fragments were mapped to hg19 using Bowtie2. Custom scripts for data processing are provided in Supplementary Software and can be downloaded from https://github.com/peteskene. For comparative analyses, publicly available datasets downloaded from the NCBI SRA archive were: ERR718799 (Abf1), SRR2568522 (Reb1), GSM749690 (CTCF; 150 bp sliding window at a 20 bp step across the genome with a false discovery rate of 1%), and the CTCF ChIP-exo BAM file was kindly provided by Frank Pugh.

## Motif identification

To obtain sets of TF-specific motifs without biasing towards CUT&RUN peaks, we applied the MEME motif-finding program to yeast ORGANIC ChIP-seq peak calls. We used the resulting log-odds position-specific scoring matrix (PSSM) for MAST searching of the *S. cerevisiae* genome to identify sites with significant log-odds motif scores. This identified 1899 Abf1 sites and 1413 Reb1 sites. Following previous studies, we use correspondence of a yeast TF binding site to the motif for that TF to be the 'gold-standard' for a true-positive call (*Rhee and Pugh, 2011*; *Kasinathan et al., 2014*; *Zentner et al., 2015*; *Ganapathi et al., 2011*). MEME was used to construct log-odds PSSMs from peaks called using the threshold method of Kasinathan et al. (*Kasinathan et al., 2014*). Peak-calling cut-off was the 99.5th percentile of normalized counts for pooled 1 s−32 s ≤120 bp Abf1 and Reb1 datasets, where the interpeak distance = 100, minimum peak width = 50, and maximum

peak width = 1000. To compare CUT&RUN and ORGANIC motif recovery, peak-call thresholds were adjusted to report similar numbers of peaks. Log-odds sequence logos were produced using PWMTools (http://ccg.vital-it.ch/pwmtools/). Track screen shots were produced using IGV (*Thorvaldsdottir et al., 2013*)

## Accession codes

Sequencing data have been deposited in GEO at National Center for Biotechnology Information under the accession number GSE84474.

## Acknowledgements

We thank Christine Codomo for preparing Illumina sequencing libraries, Jorja Henikoff for bioinformatics, Uli Laemmli for the pA-MN plasmid and for helpful discussions, Aaron Hernandez for preparing pA-MN protein, the Fred Hutch Genomics Shared Resource for sequencing, and Kami Ahmad, Siva Kasinathan and Paul Talbert for comments on the manuscript.

## Additional information

### Funding

| Funder | Grant reference number | Author |
| --- | --- | --- |
| Howard Hughes Medical Institute | Henikoff | Peter J Skene<br>Steven Henikoff |

The funders had no role in study design, data collection and interpretation, or the decision to submit the work for publication.

### Author contributions

PJS, Conceptualization, Software, Formal analysis, Validation, Investigation, Methodology, Writing—original draft, Writing—review and editing; SH, Conceptualization, Formal analysis, Funding acquisition, Validation, Investigation, Methodology, Writing—original draft, Writing—review and editing

### Author ORCIDs

Steven Henikoff, (iD) http://orcid.org/0000-0002-7621-8685

## Additional files

### Major datasets

The following dataset was generated:

| Author(s) | Year | Dataset title | Dataset URL | Database, license, and accessibility information |
| --- | --- | --- | --- | --- |
| Skene PJ, Henikoff S | 2016 | Cut-and-Run in situ factor profiling maps DNA binding and 3D contact sites at high resolution | http://www.ncbi.nlm.nih.gov/geo/query/acc.cgi?token=ynsngiuiplovju-t&acc=GSE84474 | Publicly available at the NCBI Gene Expression Omnibus (accession no: GSE84474) |

The following previously published datasets were used:

| Author(s) | Year | Dataset title | Dataset URL | Database, license, and accessibility information |
| --- | --- | --- | --- | --- |
| Kasinathan S, Orsi GA, Zentner GE, Ahmad K, Henikoff S | 2014 | High-resolution mapping of transcription factor binding sites on native chromatin. | https://www.ncbi.nlm.nih.gov/geo/query/acc.cgi?acc=GSE45672 | Publicly available at the NCBI Gene Expression Omnibus (accession no: GSE45672) |

| Tang Z, Luo OJ, Li X, Zheng M, Zhu JJ, Szalaj P, Trzaskoma P, Magalska A, Wlodarczyk J, Ruszczycki B, Michalski P, Piecuch E, Wang P, Wang D, Tian SZ, Penrad-Mobayed M, Sachs LM, Ruan X, Wei CL, Liu ET, Wilczynski GM, Plewczynski D, Li G, Ruan Y | 2015 | CTCF-Mediated Human 3D Genome Architecture Reveals Chromatin Topology for Transcription | https://www.ncbi.nlm.nih.gov/geo/query/acc.cgi?acc=GSE72816 | Publicly available at the NCBI Gene Expression Omnibus (accession no: GSE72816) |

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

## Appendix 1: Native ChIP protocol

Protocol has been shown to:

- Solubilize ~100% of Pol2/CTCF/Drosophila cenH3
- Result in no re-distribution of CTCF
- Provide high resolution maps
- Very low background (most background comes from longer DNA fragments than TF foot-prints, so is amenable to size selection to enrich for short fragments).

Solubilization:

- SDS is vital to the complete solubilization of proteins (we used a final concentration of 0.05%).
- The EDTA in the stop buffer is also beneficial in extraction.
- Verify by western blot analysis.

MNase conditions:

- If digestion is too extensive then it appears that MNase will chew through the TF footprints.
- Best results have been obtained when the input is digested to give similar stoichiometries for mono-, di- and tri-nucleosomes.
- Analyze fragments lengths 20–75 bp.

Procedure:

1. Harvest cells; spin down at 600 g 3 min in swing bucket rotor (for mammalian cells, 5–10 $\times$ $10^6$ cells are typically used per ChIP in a 1 ml MNase reaction).
2. Wash cells by resuspending in 1 ml cold PBS (scale up if greater than 10 million cells) by gently pipetting. Spin down as above.
3. Resuspend cells by gently pipetting in 1 ml cold dilution buffer (20 mM Tris-HCl pH8.1, 150 mM NaCl, 2 mM EDTA, 1% Triton X-100) supplemented with 0.05% SDS, 3 mM $CaCl_2$ and protease inhibitors).
   - Place on ice for 10 min
   - Scale up if more than 10 million cells
4. Pre-warm tubes by placing in 37°C water bath for 2 min.
5. Add pre-determined amount of MNase (Sigma N3755 @ 0.2 units per µl); incubate at 37°C
6. Stop by adding EDTA to 10 mM and EGTA to 20 mM; make a master mix to allow faster addition. Mix by inversion and place on ice immediately
7. Sonicate using a Covaris M220: 2.5 min; 10% Duty; 75 W Power; 200 cycles/burst; 7°C
8. Place on end-over-end table at 4°C for 1 hr
   - Take 20 µl as Total for protein analysis
9. Spin for 2 min at 16000 g at 4°C
   - Take 20 µl as input for protein analysis
   - Take 100 µl as input (store at 4°C overnight)
   - Take 800 µl for IP; add antibody (place on end-over-end table at 4°C overnight)
10. Next morning, prepare Protein G dynabeads, by washing 3 x with unsupplemented dilution buffer. Add 100 µl to each IP. Place on end-over-end table at 4°C for 1 hr.
11. Prepare ice-cold wash buffer: use the same the same SDS concentration as step three but _no_ $CaCl_2$
12. Do 5 $\times$ 800 µl washes (keep a fraction of the unbound for protein analysis)
13. Extract DNA by adding 300 µl extraction buffer to the beads. Place at 55°C for 1 hr

| 1 rxn | Extraction buffer |
| --- | --- |
| 6 µl | 1 M Tris pH 8.0 (final 20 mM) |
| 6 µl | 0.5 M EDTA (final 10 mM) |
| 3 µl | 0.2 M EGTA (final 5 mM) |

*continued*

| 1 rxn | Extraction buffer |
|---|---|
| 3 µl | 10% SDS (final 0.1%) |
| 18 µl | 5M NaCl (final 300 mM) |
| 5 µl | Proteinase K |
| 1 µl | RNaseA |
| 258 µl | $H_2O$ |
| 300 µl | |

14. Phenol/Chloroform extract. EtOH precipitate with glycogen. 70% EtOH wash.
15. Resuspend in 20 µl $H_2O$ (can add RNaseA to the entire prep if desired).
16. Run 3 µl Input on a 1.8% agarose-TAE gel (*RNaseA treat before loading*).

Skene and Henikoff. eLife 2017;6:e21856. DOI: 10.7554/eLife.21856

# Appendix 2: CUT&RUN protocol for yeast

## Nuclei

- From ~$5 \times 10^8$ *S. cerevisiae* cells @ OD$_{600}$ ~0.7 (*Orsi et al., 2015*).
- Other methods for preparing nuclei are expected to give equivalent results.
- Bio-Mag Plus Concanavalin A coated beads can be purchased from Polysciences, Inc. (Warrington, PA, Catalog #86057).

## Digestion buffer (150 ml)

| | |
|---|---|
| 3 ml 1M HEPES pH 7.5 | 20 mM |
| 3 ml 5M NaCl | 100 mM |
| water to 150 ml | |

- Add 1 mM phenylmethanylsulfonyl fluoride (PMSF, 100 mM stock in ethanol) just before use and hold on ice after addition.

## HNT wash buffer (100 ml)

| | |
|---|---|
| 100 ml Digestion buffer | |
| 100 µL Tween 20 | 0.1% |
| 1 mM PMSF just before use (= HNT-Pi) | |

## For preparing beads

- HNT ++ = HNT +1 mM CaCl$_2$, + 1 mM MnCl$_2$
- Ca$^{++}$ and Mn$^{++}$ are needed to activate lectins. Although the manufacturer recommends 1 mM MgCl$_2$ as well, this can cause DNA degradation and omitting MgCl$_2$ had no effect on binding of permeabilized nuclei to beads.

## HNT-PPi blocking buffer (20 ml):

| | |
|---|---|
| 20 ml HNT wash buffer | |
| 67 µL 30% BSA | 0.1% |
| 2 mini-Complete Ultra (Roche) protease inhibitor tablets | |
| 1 mM PMSF just before use | |

## HNT-preblock (per 1 ml)

| | |
|---|---|
| 900 µL HNT-PPi | |
| 100 µL 30% BSA | 3% |
| 4 µL 0.5M EDTA | 2 mM |
| 1 mM PMSF just before use | |

## 2XSTOP (10 ml)

| | |
|---|---|
| 400 µl 5M NaCl | 200 mM |
| 400 µL 0.5M EDTA | 20 mM |
| 200 µL 0.2M EGTA | 4 mM |
| + 50 µL Thermo RNase A (10 mg/ml) | 50 µg/ml |
| + 20 µL glycogen (20 mg/ml) | 40 µg/ml |
| water to 10 ml | |

- For spike-in add ~10 pg/ml spike-in DNA (*e.g.* mononucleosome-sized fragments from MNase digestion of formaldehyde crosslinked Drosophila S2 cells).

## 1XSTOP (10 ml)

| | |
|---|---|
| 200 µl 5M NaCl | 100 mM |
| 200 µL 0.5M EDTA | 10 mM |
| 100 µL 0.2M EGTA | 2 mM |
| water to 10 ml | |

## Procedure

- Pre-blocking: Add 1 ml HNT-preblock with gentle pipetting. Let sit 5 min, then spin, place on the magnet stand, pull off the supernatant and continue to the next step.
- Antibody binding: Block 5 min in 1 ml blocking buffer (HNT-PPi with 0.1% BSA). Place on the magnet stand and pull off the supernatant and bring up in 500 µL blocking buffer. While vortexing gently add 500 µL anti-FLAG (containing 5 µL Sigma M2 mouse anti-FLAG antibody – 1:200 final). Incubate on rotator 2 hr at 4°C. Spin and wash once in 1 ml HNT-Pi Wash buffer.
- Secondary antibody binding (optional): If a mouse monoclonal antibody was used, a rabbit anti-mouse secondary antibody (*e.g.* Abcam ab46540) is needed to provide high specificity of pA-MN binding. Use of a secondary antibody amplifies the cleavage rate by 1–2 orders of magnitude. Follow the same procedure as in Step 3, except incubate for 1 hr.
- Bind pA-MN: Pull off the supernatant and bring up in 500 µL blocking buffer. While vortexing add 500 µL of blocking buffer containing 5 µL pA-MN (600 µg/ml). Incubate 1 hr on rotator at 4°C. Spin and wash twice in 1 ml HNT-Pi Wash buffer.
- Digestion: Decant and wash once in 1 ml Digestion buffer, bring up in 1.2 ml Digestion buffer, and divide each into 8 150 µL time-point aliquots, placing directly on the bottom of the tube. Equilibrate to 0°C on blocks fitted for 1.7 ml tubes in ice-water. Place a 3 µL 100 mM CaCl$_2$ (to 2 mM) drop on the side of each tube. To obtain a time course, begin digestion by vortexing tubes and replacing in the ice-water holder. Stop by addition of 150 µL 2XSTOP (optionally with spike-in DNA added).
  For total DNA extraction: Add 3 µL 10% SDS (to 0.1%), and 2.5 µL Proteinase K (20 mg/ml) to samples and vortex. Incubate at 70°C for 10 min with occasional inversion to mix. (For formaldehyde crosslinked cells incubate ≥4 hr at 65°C to reverse the crosslinks.) Mix with 300 µL phenol-chloroform-isoamyl, spin 5 min 13 krpm and decant to a fresh tube. Add ½ volume (150 µL) AMPure beads and mix well. Let sit 10 min and place on a magnet stand. Transfer the supernatant to a fresh tube to remove remaining beads, then precipitate the supernatant with 1 ml ethanol, chill and spin. Wash in 1 ml 100% ethanol and bring up in 25 µL 0.1xTE8 for library preparation.

> For chromatin-associated complexes: Follow the total DNA extraction procedure. Include 0.5 mM spermidine in the HMT, Digestion and STOP buffers.
>
> For salt fractionation: Reduce the volume of the digestion slurry from 150 µL to 50 µL and stop the reaction with 2XSTOP where 4M NaCl is substituted for 200 mM NaCl, and leave out the RNase. After incubation at 37°C add 200 µL RNase (100 µg/ml) in water, incubate 20 min at 37°C, then continue with the 5' 13,000 rpm spin to separate the supernatant from the pellet.

- Isolating excised fragments: Incubate 37°C 20 min. Spin 5' 13,000 rpm 4°C, place on magnet stand and pull off supernatants to fresh tubes. Bring the bead pellet up in 300 µL 1XSTOP (no RNase or glycogen). Add 3 µL 10% SDS (to 0.1%), vortex, and add 2.5 µL Proteinase K (20 mg/ml) to samples. Incubate at 70°C for 10 min with occasional inversion to mix.

- Extract supernatant DNA for libraries: Mix with 300 µL phenol-chloroform -isoamyl alcohol, transfer to phase-lock tubes, spin, then extract with 300 µL chloroform. Remove to a fresh tube containing 2 µL of 2 mg/ml glycogen in the tube before addition. Add 750 µL ethanol, chill and spin. Wash the pellet in 1 ml 100% ethanol, air dry and dissolve in 25 µL 0.1xTE8. Some of the DNA represents on the order of ~1% of the high molecular weight DNA that becomes solubilized, but which will not appreciably amplify during library preparation.

- Extract pellet fraction for gel analysis (optional): Mix with 300 µL phenol-chloroform-isoamyl alcohol, spin 5 min 13,000 rpm, put on a magnet stand for ~5 min and pull off. Remove aqueous layer to a fresh tube containing 2 µL of 2 mg/ml glycogen in the tube before addition. Add 750 µL ethanol, chill and spin. Wash the pellet in 1 ml 100% ethanol, air dry and dissolve in 25 µL 0.1xTE8, then centrifuge 10 min 13 krpm to pellet most of the insoluble brown material that came off the beads.

# Appendix 3: CUT&RUN protocol for mammalian cells

Protocol has been used to map CTCF, Myc, Max and H3K27me3 in human K562 cells. We see *very* low background. This protocol relies on the cut chromatin fragments 'leaching' out of the intact nuclei into the reaction volume. The intact nuclei are spun down at the end of the experiment and the DNA extracted from the supernatant fraction. This isolates the liberated chromatin fragments and therefore does not need further size selection. The protocol can either use centrifugation (600 g; 3 min; swing-bucket rotor) or concanavalin A coated magnetic beads (BioMag Plus #86057) to isolate nuclei at each step.

Typical experimental samples: ($10 \times 10^6$ cells per reaction)

i.   no antibody; free pA-MNase (i.e. PA-MNase not washed out).
ii.  no antibody + pA-MNase (controls for background MNase activity)
iii. antibody + pA-MNase (experimental sample)

We take small QC samples before $CaCl_2$ addition ('input') and after the reaction has been stopped ('end') to assay how the MNase reaction proceeded prior to fractionation.

Protease inhibitors (Roche complete EDTA-free) are added to buffers at a final concentration of 1x from 50x stock.

1. *Optional:* Prepare beads (use 50 µl beads per $10 \times 10^6$)

• Wash 3 times in 3 volumes of binding buffer
• Resuspend in 1 volume binding buffer

2. Harvest cells; spin down at 600 g 3 min in swing bucket rotor (typically 10 million cells per sample).

3. Wash cells by resuspending in 1 ml cold Phosphate Buffered Saline (scale up if greater than 10 million cells) by gently pipetting. Spin down as above.

4. Resuspend cells in 1 ml NE1 by gently pipetting (scale up if greater than 10 million cells)

• Place on ice for 10 min

Magnetic beads:

• Spin down as above and resuspend in NE1
• Add beads directly to resuspended in nuclei, with gentle pipetting
• 5 min @ room temp on mixing platform
• bind to magnet for ~2 min, discard supernatant

or

Centrifugation:

• pellet nuclei at 600 g 3 min in swing bucket rotor

5. Resuspend in 1.7 ml CUT&RUNBuffer 1 by gently pipetting and transfer to 1.7 ml Eppendorf tubes.

• Place on ice for 5 min
• Collect nuclei by magnet or centrifugation as above

6. Resuspend in 1.5 ml CUT&RUN Buffer 2 by gently pipetting.

• Collect nuclei by magnet or centrifugation as above

7. Resuspend in CUT&RUN Buffer 2 by gently pipetting.

- Use 10 million cells in 500 μl volume in 0.5 ml Eppendorf tubes
- Add antibody as required including secondary antibody
- Place on mixing platform at 4°C for 2 hr

(0.5 ml tubes give a tighter pellet for centrifugation and reduce the sloshing of liquid during the incubation to maintain nuclear integrity)

8. Three 5 min washes with 500 μl CUT&RUN Buffer 2 on a mixing platform at 4°C.

- Collect nuclei by magnet or centrifugation as above

9. Resuspend in 300 μl CUT&RUN Buffer 2.

- Add 3 μg protein A-MNase fusion (5 μl @ 600 ng/μl _or_ 8.3 μl @ 360 ng/μl)
- Place on mixing platform at 4°C for 1 hr

(300 μl reaction volume allows the supernatant fraction to be easily extracted/EtOH ppt in a 1.7 ml tube)

10. Three 5 min washes with 300 μl CUT&RUN Buffer 2 on a mixing platform at 4°C.

- _NOT_ for free MNase sample (keep on mixing platform)
- Collect nuclei by magnet or centrifugation as above

11. Resuspend in 300 μl CUT&RUN Buffer 2.

- Take 12 μl as 'input' and place into 288 μL DNA extraction buffer

12. Place tubes in wet ice (_it is imperative the digestion is performed at 0°C – preferably use an aluminum block to maintain the temperature_).

- Add $CaCl_2$ to final concentration of 2 mM (6 μl of 100 mM $CaCl_2$)
- Mix rapidly by inverting and place on wet ice
- Incubate for desired time (e.g. 15 min)
- Typically place free MNase sample at 37°C for 5 min. This allows digestion to be evaluated by agarose gel electrophoresis.

13. Stop by adding a master mix of EDTA to 10 mM and EGTA to 20 mM

- Mix rapidly by inverting and place on ice
- Take 12 μl as 'end' and place into 288 μL DNA extraction buffer
- _Option:_ add spike-in DNA

## Option A

14. Place on a mixing platform at 4°C for 1 hr to let the chromatin fragments leach out

- Spin down at 600 g 3 min SW rotor (_even if using the magnetic bead approach_)
- Take supernatant

15. Extract DNA from supernatant by adding:

- 3 μl 10% SDS (final concentration 0.1%)
- 5 μl proteinase K at 10 mg/ml
- 2 μl RNaseA at 1 mg/ml
- 5 μl 5 M NaCl (final concentration 300 mM)
- Vortex and place at 55°C for 1 hr
- Phenol extract; EtOH precipitate (add 1 μl glycogen); EtOH wash
- Resuspend in 20 μl $H_2O$

## Option B

14. Other option is to extract all DNA and then use a very simple size selection to separate the large uncut pieces of the genome from the small footprints. This is a safer bet for large, potentially insoluble protein complexes that might not diffuse through the nuclear pores.

### Extract DNA from entire reaction:

- 3 µl 10% SDS (final concentration 0.1%)
- 5 µl proteinase K at 10 mg/ml
- 2 µl RNaseA at 1 mg/ml
- 5 µl 5 M NaCl (final concentration 300 mM)
- Vortex and place at 55°C for 1 hr
- Phenol extract; EtOH precipitate (add 1 µl glycogen); EtOH wash
- Resuspend in 150 µl $H_2O$

15. Size selection of cut fragments ($\leq \sim$ 700 bp) using Beckmann Agencourt AMPure XP beads (A63881)

- Allow beads to warm up to room temp before use
- Add 75 µl beads, mix by pipetting 10x
- Incubate at room temp for 5 min
- Place on magnet for 2 min
- Take supernatant fraction (it is imperative not to take any of high MW DNA attached to the beads, one can spin down the supernatant fraction to check for beads)
- precipitate by adding 700 µl EtOH and 1 µl glycogen (no additional salt required)
- 70% EtOH wash
- Resuspend in 20 µl $H_2O$

## Buffers:

Protease inhibitors (Roche complete EDTA-free) added to 1x from 50x stock in water

Binding buffer

- 1 x PBS
- 1 mM $CaCl_2$
- 1 mM $MgCl_2$
- 1 mM $MnCl_2$

NB: We have found that nucleases in some cells cause $Mg^{++}$-dependent degradation of DNA. The presence of $Mg^{++}$ in the binding buffer follows the manufacturer's recommendation, but only $Ca^{++}$ and $Mn^{++}$ are needed to activate lectins. Omitting $MgCl_2$ had no effect on binding of permeabilized cells to beads.

NE1:

- 20 mM Hepes-KOH pH 7.9
- 10 mM KCl
- 1 mM $MgCl_2$
- 0.1% Triton X-100
- 20% Glycerol

NB: We find that substituting 0.5 mM spermidine for 1 mM $MgCl_2$ can be used to avoid $Mg^{++}$-dependent DNA degradation.

CUT&RUN Buffer 1:

- 20 mM Hepes pH 7.5
- 150 mM NaCl
- 2 mM EDTA

- 0.5 mM Spermidine
- 0.1% BSA

CUT&RUN Buffer 2:

- 20 mM Hepes pH 7.5
- 150 mM NaCl
- 0.5 mM Spermidine
- 0.1% BSA

Other reagents:

- 100 mM $CaCl_2$
- 10% SDS
- 5M NaCl
- 500 mM EDTA
- Proteinase K
- 500 mM EGTA
- RNaseA
- Extraction buffer (See Appendix 1)

## DNA extraction for 12 µl QC samples taken during protocol:

- Phenol extract
- Ethanol precipitate
- Ethanol wash
- Resuspend in 20 µL $H_2O$
- RNase-treat
- Electrophorese on 0.7% agarose gel

