## [Decision Letter]

Thank you for submitting your article "CUT&RUN: An efficient alternative strategy for high-resolution mapping of DNA binding sites" for consideration by *eLife*. Your article has been favorably evaluated by Jessica Tyler (Senior Editor) and three reviewers, one of whom is a member of our Board of Reviewing Editors. The reviewers have opted to remain anonymous.

The reviewers have discussed the reviews with one another and the Reviewing Editor has drafted this decision to help you prepare a revised submission.

The method described is interesting and compared with conventional ChIP-seq results in higher resolution, lower background, and greater versatility, where mapping of long range interaction showed near base-pair resolution. A limitation/concern is that the method was used to analyze the binding of CTCF (and the centromeric histone variant CenH3). We request that the authors analyze another factor with limited binding sites in the genome such as a pioneer factor that binds to inaccessible, thus regions resistant to light MNAse treatment. Additionally, an explanation as to how the authors can tell direct from indirect cuts due to topology is necessary.

*Reviewer #1:*

In the manuscript by Skene and Henikoff, the authors developed Cleavage Under Targets and Release Using Nuclease (CUT & RUN), a chromatin immunocleavage (ChIC) based genome-wide mapping strategy. Compared with conventional ChIP-seq, this new method results in higher resolution, lower background, and greater versatility. In this method, native nuclei were immobilized and incubated with antibody against the protein of interest, and treated with protein A-MNase fusion protein for a very short time at 4o C. DNA released from MNase digestion was subjected to deep-sequencing. The authors tested this method with transcription factors including CTCF and with centromeric histone variant CenH3, and reported data with high resolution and sensitivity. Subsequent long-range interaction mapping by this method showed near base-pair resolution. Overall, the study provided a useful option for genome-wide mapping. There are few points that need to be addressed:

1) It could be problematic to use budding yeast to study insoluble DNA-binding protein complexes. Yeast is a relatively simple model system, and more importantly, most of the yeast genome is actively transcribed, which is very different from the mammalian genome. The mammalian genome is more complex and has large regions of heterochromatin. It is not clear if those regions are accessible to the CUT & RUN methodology. The authors should test the applicability of their methodology to mammalian cells. At the very least, the authors must state in the text the potential limitations of this new methodology.

2) It is unclear if the method can be used to probe chromatin states, including a variety of histone modifications. These features are very topical in the field of chromatin biology. Most current genome-wide profiling approaches can provide useful information on chromatin modifications. It would be useful for the authors to test if CUT & RUN can provide data on genome-wide histone modifications. If not, the authors should discuss this deficiency.

3) In the second paragraph of the subsection CUT&RUN robustly maps yeast TF binding sites in situ at high resolution”, how did the authors define "true positives for both TFs"?

*Reviewer #2:*

The authors describe a technique for quantitative genome-wide mapping of chromatin-bound factors using antibodies and MNase in nucleo, without requiring fixation or sonication. They show efficient and accurate mapping of two yeast transcription factors at high resolution, precise mapping of the centromeric nucleosome, which is rare in abundance and trapped in an insoluble complex, and mapping of CTCF in human cells, including direct binding sites as well as indirect 3D contact points.

The technique is ingenious and the study is very well presented. I particularly want to applaud the authors for including clear and detailed supplementary protocols, which is rarely seen. If the technology worked as well as claimed and were a true alternative for ChIP-seq, I would highly recommend this manuscript for publication. However, I have a major concern about the ability of CUT&RUN to distinguish direct from indirect contacts and additional worries about specificity and versatility that would need to be addressed experimentally.

A) 3D contacts: the authors present the ability of CUT&RUN to identify distal indirect contacts of CTCF as an asset, but to me it appears to be a major limitation. If I understand correctly, of the ~20k sites identified by CUT&RUN, only those showing native ChIP signal (~10%) are direct binding sites; the others are indirect. That would imply that native ChIP must be performed in parallel to CUT&RUN to identify direct binding sites. That seem a major limitation that disqualifies the technique as a full replacement for ChIP-seq. Is this limited to CTCF or they see evidence of indirect sites also in their yeast data? Are these indirect contacts captured by conventional X-ChIP? Is there any way to distinguish direct from indirect bioinformatically or by modifications to the CUT&RUN protocol?

B) Specificity: the authors show convincingly that CUT&RUN is as sensitive or more sensitive than IP methods. They also show that the background noise is lower (e.g. Figure 1) than in ChIP-seq. However, this is a relatively innocuous type of background as it is uniformly low. A more important question is if CUT&RUN identifies false positive peaks in accessible chromatin regions, as some of the CTCF data seems to suggest. The authors partially deal with this in Figure 5—figure supplement 1, but I believe that it is crucial to address this more thoroughly. Specifically I suggest that:

B1) the authors perform CUT&RUN experiments in cells lacking the targeted protein.

B2) the authors perform detailed analyses to determine false positive rates, including calling peaks for Abf1 and Reb1 and showing that the majority of the top peaks are within motifs.

B3) the authors analyze enrichment over DNaseI hypersensitive sites for all the CUT&RUN experiments and compare it with enrichment in native ChIP and X-ChIP as a measure of false positive rate.

C) Versatility: to claim that CUT&RUN is a "suitable replacement" (Abstract) or even just "an attractive alternative" (Discussion) to ChIP, the authors should show that it can be applied to a similarly broad array of chromatin-binding proteins. Specifically, my concern is that CUT&RUN works best when the targeted protein resides in a nucleosome free region, and the three proteins targeted here belong to that category. The experiment on Cse4 helps but a more broadly distributed heterochromatic protein would be more convincing.

D) Scales: in several points it is hard to follow the comparisons because scales are omitted or vague (e.g. "low" to "high") in genome browser snapshots and heatmaps. They must be included for all plots and the values represented fully described in the text or legends.

*Reviewer #3:*

The authors build on their previously developed method to dig deeper into the possibilities offered by ChIP variants to precisely map the genomic location of DNA binding proteins. The technique has the potential to be scalable and transferable and thus impactful.

As the authors point out, the preferred method to analyze genomic occupancy is ChIP-seq because of its relatively simplicity and adaptability to different proteins. Other methods that tag specific enzymes to binding sites are inherently more tedious to apply to multiple proteins because they require the construction of protein fusions. Additionally, they sometimes lack resolution or depth. However, they have the advantage of working in native conditions or even in vivo. The presented method has the advantages of ChIP-seq since it requires only an antibody to the protein of interest without the need of generating a transgenic cell lines. Thus, it can be complemented and compared directly to ChIP-seq. It also gains in resolution over conventional ChIP-seq by relying on an enzymatic digestion of the DNA under native conditions. Additionally, by releasing MNase accessible regions, the method reduced background reads. Finally, it allows the investigation of the local accessibility state. Thus, with minor controls it can be a valuable method for the large community studying how specific proteins interact with the genome.

The described robust performance across digestions time is a very positive attribute of this protocol. Other protocols that require MNase digestion are extremely sensitive to its activity and thus very hard to set up. The overlap on size distribution of Figure 1 demonstrate that fragments are of the same size but it is not conclusive for further applications in terms of identity of these fragments. If the size distribution is similar but globally these fragments map to different locations in the genome, the timing on digestion is an extremely important parameter. Although Figure 2—figure supplement 1 attempts to map it, a global mapping and overlap quantification of the data represented for a region in Figure 1 would benefit the understanding of the digestion time importance or lack of it.

The authors compare Cut&Run with ORGANIC ChIP-seq, a previous generation method from their latest improved method. How does this method compare to their latest method to map TF binding at single base-pair resolution published last year?

Cut&Run relies on MNase digestion. How does Cut&Run perform for pioneer transcription factors that bind to inaccessible, thus resistant regions to light MNAse treatment? Thus, should be discussed and perhaps tested (see below).

Related, and in light of this comment "When aligned to CTCF motifs found within DNaseI hypersensitive 245 sites, CUT&RUN and X-ChIP-seq CTCF heat maps show strong concordance": How does it compare to whole genome ChIP-seq? How does the >150bs run compares to an accessibility experiments? Would a MNase experiment be a better background model for peak calling? If true, then the impact of this method should be reworded.

CTCF is an atypical TFs in terms of its ability to ChIP. Few novel protocols have been established and tested with CTCF but fail when repeated with other TFs. Thus, I highly recommend to perform and report a Cut&Run experiment with another TF such as homeodomain. This to me is an important and the only wet experiment the authors should perform. The rest of my comments could be addressed by re-analysis of the data.

The claim about recovering 3D interactions is weak and to me there is not enough support to include this claim in the manuscript. First, it needs the MNase background model. The MNase is not covalently bound to CTCF, thus some of the release fragments could be released at similar frequencies in both conditions, while CTCF-interacting elements will be resealed at higher frequencies. Secondly, no technique is perfect, but discussion about transient binding and its implications for the digestion of certain regions should be suggested. Finally, the most important information from 3C techniques is that they identify the binding partners. Cut&Run even in the best case scenario will identify regions that could be in contact with one of the many CTCF binding sites in the genome.

---

## [Author Response]

*The method described is interesting and compared with conventional ChIP-seq results in higher resolution, lower background, and greater versatility, where mapping of long range interaction showed near base-pair resolution. A limitation/concern is that the method was used to analyze the binding of CTCF (and the centromeric histone variant CenH3). We request that the authors analyze another factor with limited binding sites in the genome such as a pioneer factor that binds to inaccessible, thus regions resistant to light MNAse treatment. Additionally, an explanation as to how the authors can tell direct from indirect cuts due to topology is necessary.*

We thank the reviewers and editors for their enthusiasm for the method and for their many helpful comments and suggestions, which have greatly improved the manuscript. We now provide new data and analyses showing that CUT&RUN performs better than conventional ChIP-seq for a broad range of other factors, now directly addressing issues of DNA accessibility, background and versatility. As for the second issue, we now show that direct and indirect site detection is the same for CUT&RUN and X-ChIP-seq, and that in either case, a method such as our new native ChIP protocol is needed to decide. We have also addressed each specific comment in our point-by-point response.

*Reviewer #1:*

*[…]*

*1) It could be problematic to use budding yeast to study insoluble DNA-binding protein complexes. Yeast is a relatively simple model system, and more importantly, most of the yeast genome is actively transcribed, which is very different from the mammalian genome. The mammalian genome is more complex and has large regions of heterochromatin. It is not clear if those regions are accessible to the CUT & RUN methodology. The authors should test the applicability of their methodology to mammalian cells. At the very least, the authors must state in the text the potential limitations of this new methodology.*

We agree that the concern that tethered MNase cannot break through mammalian heterochromatin needs to be addressed. Accordingly, we have performed CUT&RUN using an antibody against H3K27me3, which is a marker for facultative heterochromatin. The resulting profiles match ChIP-seq profiles by the ENCODE project with much better dynamic range using the same number of reads (new Figure 6—figure supplement 4, subsection "CUT&RUN maps histone modifications in compacted chromatin”).

*2) It is unclear if the method can be used to probe chromatin states, including a variety of histone modifications. These features are very topical in the field of chromatin biology. Most current genome-wide profiling approaches can provide useful information on chromatin modifications. It would be useful for the authors to test if CUT & RUN can provide data on genome-wide histone modifications. If not, the authors should discuss this deficiency.*

We agree that users of CUT&RUN will want to apply the method to histone modifications. As explained in response to point 1, we have addressed this concern by showing that CUT&RUN provides H3K27me3 profiles that are better than the current standard: ChIP-seq datasets published by the ENCODE project. Therefore, CUT&RUN provides an attractive alternative to ChIP for histone modifications.

*3) In the second paragraph of the subsection "CUT&RUN robustly maps yeast TF binding sites in situ at high resolution”, how did the authors define "true positives for both TFs"?*

We now explicitly define true positives in a “Motif identification” subsection in Methods as sites with statistically significant motifs determined by MAST searching of the yeast genome, using motifs found by MEME with ChIP-seq and CUT&RUN peak calls. The close correspondence between motifs determined using both CUT&RUN and ChIP-seq (Figure 1—figure supplement 4) provides additional confidence in the use of the motif determined statistically as the “gold standard” for identifying true positives, which is now mentioned in the text.

*Reviewer #2:*

*The authors describe a technique for quantitative genome-wide mapping of chromatin-bound factors using antibodies and MNase in nucleo, without requiring fixation or sonication. They show efficient and accurate mapping of two yeast transcription factors at high resolution, precise mapping of the centromeric nucleosome, which is rare in abundance and trapped in an insoluble complex, and mapping of CTCF in human cells, including direct binding sites as well as indirect 3D contact points.*

*The technique is ingenious and the study is very well presented. I particularly want to applaud the authors for including clear and detailed supplementary protocols, which is rarely seen. If the technology worked as well as claimed and were a true alternative for ChIP-seq, I would highly recommend this manuscript for publication. However, I have a major concern about the ability of CUT&RUN to distinguish direct from indirect contacts and additional worries about specificity and versatility that would need to be addressed experimentally.*

We thank Reviewer 2 for raising these important issues, which we now address with additional data, analyses and clarifications.

*A) 3D contacts: the authors present the ability of CUT&RUN to identify distal indirect contacts of CTCF as an asset, but to me it appears to be a major limitation. If I understand correctly, of the ~20k sites identified by CUT&RUN, only those showing native ChIP signal (~10%) are direct binding sites; the others are indirect. That would imply that native ChIP must be performed in parallel to CUT&RUN to identify direct binding sites. That seem a major limitation that disqualifies the technique as a full replacement for ChIP-seq. Is this limited to CTCF or they see evidence of indirect sites also in their yeast data? Are these indirect contacts captured by conventional X-ChIP?*

Yes, the inability to distinguish direct from indirect sites is common to both X-ChIP-seq and CUT&RUN. The evidence for this is presented in the heat maps in the new Figure 6—figure supplement 1, where essentially all of the CUT&RUN calls have ENCODE calls. However, we had failed to make this point clear in the text, and we thank reviewer 2 for pointing out this important omission. To clarify this point we have now discussed this correspondence where we describe the indirect sites (“CUT&RUN maps long-range genomic contacts”). To further emphasize the fact that all X-ChIP methods capture both direct and indirect sites, we have added a heat map for our published CTCF MNase X-ChIP-seq sites, which also shows that X-ChIP-seq peaks and CUT&RUN peaks correspond closely (Figure 6). As for our yeast data, by using the consensus motif as a gold standard to distinguish true positives from false positives, the very low level of false positive calls we see for Abf1 and Reb1 using both CUT&RUN and ORGANIC argues that strong indirect sites for these TFs are rare at best.

*Is there any way to distinguish direct from indirect bioinformatically or by modifications to the CUT&RUN protocol?*

We have not found an experimental way to separate the long-range cleavages from those nearby, as they occur nearly simultaneously during the digestion time-course.

*B) Specificity: the authors show convincingly that CUT&RUN is as sensitive or more sensitive than IP methods. They also show that the background noise is lower (e.g. Figure 1) than in ChIP-seq. However, this is a relatively innocuous type of background as it is uniformly low. A more important question is if CUT&RUN identifies false positive peaks in accessible chromatin regions, as some of the CTCF data seems to suggest.*

As detailed below, we now show that there is no bias towards accessible chromatin regions. The peaks referred to are instead attributable to indirect contact sites, which are also present in X-ChIP-seq but not native datasets, as explained above.

*The authors partially deal with this in Figure 5—figure supplement 1, but I believe that it is crucial to address this more thoroughly. Specifically I suggest that:*

B1) the authors perform CUT&RUN experiments in cells lacking the targeted protein.

We thank reviewer 2 for raising this critical issue, which we had not adequately addressed. Nearly all of the proteins that we profiled in this study are essential, so this exact control is problematic at best. However, the underlying concern raised is that accessible chromatin regions might be a source of false positive peaks, and we address this issue with a rigorous negative control: In our yeast experiments we leave out the primary anti-FLAG antibody thus identically treating an aliquot of the same nuclei with IgG (rabbit anti-mouse) and pA-MN in parallel. After maximum digestion time for an experiment, we recover low levels of background cleavage fragments from throughout the genome. In this way, the cells are not disrupted, and the single negative control serves for all yeast proteins in the study. We now include representative negative control tracks in Figure 1 and new Figure 4—figure supplement 2, which show only noise. To directly address the question of background at accessible chromatin regions, we now show that our no primary antibody control gives a flat line through the nucleosome depleted regions (NDRs) of yeast genes lined up over the +1 nucleosome, without a hint of any DNA accessibility preference (new Figure 4 panel B). We now also make this point in the new subsection describing the new set of chromatin remodeler experiments (“CUT&RUN precisely maps chromatin-associated complexes”) and the new paragraph describing the Myc and Max CUT&RUN experiments (subsection “CUT&RUN maps human transcription factor binding sites at high resolution”, last paragraph). For human K562 cells, a no antibody control also shows that there are no preferential cleavages over NDRs found at TF binding sites (New Figure 6—figure supplement 1).

B2) the authors perform detailed analyses to determine false positive rates, including calling peaks for Abf1 and Reb1 and showing that the majority of the top peaks are within motifs.

We have now performed this analysis for Abf1 and Reb1. To compare CUT&RUN and ORGANIC motif recovery, peak-call thresholds were adjusted to report similar numbers of peaks. We find that the majority of peaks called using a threshold algorithm have a significant motif for both CUT&RUN and ORGANIC, and the relative performance is similar for both a stringent threshold (~650 peaks called) and a relaxed threshold (~1100 peaks called). By calling peaks using a simple threshold algorithm that makes no assumptions concerning peak features, and varying the threshold for high and low stringency, we believe that this performance comparison is robust. Indeed, consistent with the differences in sensitivity and specificity seen in the occupancy heat maps displayed in Figure 2, peak-calling performance was somewhat better for CUT&RUN than ORGANIC for Abf1, and vice versa for Reb1 (New Figure 1—figure supplement 4 and subsection “CUT&RUN robustly maps yeast TF binding sites in situ at high resolution”).

*B3) the authors analyze enrichment over DNaseI hypersensitive sites for all the CUT&RUN experiments and compare it with enrichment in native ChIP and X-ChIP as a measure of false positive rate.*

As pointed out above, the concern about accessible DNA is addressed with our no primary antibody negative controls both for yeast and for human. As for detection of false positives that might arise for any reason, we find that a sensitive criterion is to ask whether an Abf1 peak is found at a Reb1 site and vice-versa, such as is shown graphically in Figure 2 and now quantified as described in response to point B2. For yeast, the true positive rate is best determined by motif analysis as described above, and this has been the gold standard, at least for yeast, where we consider sites of binding without a motif to be putative false positives.

*C) Versatility: to claim that CUT&RUN is a "suitable replacement" (Abstract) or even just "an attractive alternative" (Discussion) to ChIP, the authors should show that it can be applied to a similarly broad array of chromatin-binding proteins. Specifically, my concern is that CUT&RUN works best when the targeted protein resides in a nucleosome free region, and the three proteins targeted here belong to that category. The experiment on Cse4 helps but a more broadly distributed heterochromatic protein would be more convincing.*

We agree that it is important to demonstrate versatility beyond transcription factors and nucleosomes, and we have addressed this in two ways. First, as described in response to reviewer 1, our addition of H3K27me3 shows that CUT&RUN outperforms ChIP-seq for heterochromatin. Second, we now address the question of whether CUT&RUN can profile dynamic components of the chromatin landscape. Myc and Max are considered to be especially dynamic TFs, and as explained below and shown in the new Figure 6—figure supplement 3, we obtain high-resolution Myc and Max CUT&RUN profiles with higher dynamic range than obtained by ENCODE. ATP-dependent chromatin remodelers are especially challenging for ChIP because of their mobility and large size. Despite these issues, we have found that this class of chromatin-associated proteins can be mapped precisely and efficiently by CUT&RUN, described in a new subsection (“CUT&RUN precisely maps chromatin-associated complexes”) and new figures (Figure 4 and Figure 4—figure supplement 1 and Figure 4—figure supplement 2). Data are presented for two remodelers: Mot1, the DNA translocase that evicts TATA-binding protein (TBP) from the upstream side, and RSC, the 17-subunit ~1 megadalton complex that slides nucleosomes to create nucleosome depleted regions. We find that a slightly modified CUT&RUN protocol (subsection “CUT&RUN precisely maps chromatin-associated complexes”) provides results that are similar to ORGANIC profiling of these mobile components of the chromatin landscape with only ~15% of the number of paired-end reads.

*D) Scales: in several points it is hard to follow the comparisons because scales are omitted or vague (e.g. "low" to "high") in genome browser snapshots and heatmaps. They must be included for all plots and the values represented fully described in the text or legends.*

We have corrected this issue throughout the manuscript, indicating the pixel values for the heat maps, and for the browser snapshots, we indicate the Y-axis values in the figure where peak autoscaling was done, and in the legend where the tracks in each panel are identically scaled.

*Reviewer #3:*

*The authors build on their previously developed method to dig deeper into the possibilities offered by ChIP variants to precisely map the genomic location of DNA binding proteins. The technique has the potential to be scalable and transferable and thus impactful.*

*As the authors point out, the preferred method to analyze genomic occupancy is ChIP-seq because of its relatively simplicity and adaptability to different proteins. Other methods that tag specific enzymes to binding sites are inherently more tedious to apply to multiple proteins because they require the construction of protein fusions. Additionally, they sometimes lack resolution or depth. However, they have the advantage of working in native conditions or even* in vivo*. The presented method has the advantages of ChIP-seq since it requires only an antibody to the protein of interest without the need of generating a transgenic cell lines. Thus, it can be complemented and compared directly to ChIP-seq. It also gains in resolution over conventional ChIP-seq by relying on an enzymatic digestion of the DNA under native conditions. Additionally, by releasing MNase accessible regions, the method reduced background reads. Finally, it allows the investigation of the local accessibility state. Thus, with minor controls it can be a valuable method for the large community studying how specific proteins interact with the genome.*

We thank reviewer 3 for these positive comments, but there appears to be a misunderstanding concerning the supposed use of our method to investigate local accessibility states. Here and below, reviewer 3 seems to be assuming that tethered MNase will have the same biases as free MNase, but our data show that this is not the case at all. This was a point that we made in describing how yeast kinetochores, the cores of which are the most AT-rich regions in the genome, remain fully intact in our Cse4 and H2A CUT&RUN profiles. However, the well-known preference for cleaving AT-rich DNA (e.g. McGhee & Felsenfeld, “Another potential artifact in the study of nucleosome phasing by chromatin digestion with micrococcal nuclease”, Cell 1983, PMID: 6301684) has led to the understandable assumption that tethered MNase would have the same DNA accessibility artifact. In stark contrast to this expectation, our study shows that CUT&RUN has remarkably little bias either for or against accessible or inaccessible regions genome-wide. We now demonstrate this fact more directly by showing that our No Antibody negative controls show no difference between NDRs and flanking nucleosomes for both yeast (new Figure 4) and human (new Figure 6—figure supplement 1). Also, our new H3K27me3 CUT&RUN profiling data also indicate that local accessibility is not a factor when the MNase is tethered. This is in contrast to freely diffusing MNase, which has such a strong preference for linker DNA that it has been used for 40 years to define linker accessibility (e.g. Reeves & Jones, Science 1976, PMID:1264202). As targeting MNase is the key feature that distinguishes CUT&RUN from Native ChIP-seq, MNase-seq, DNAse-seq, ATAC-seq and other popular nuclease-based methods we have modified the title to better emphasize that the MNase is *targeted*.

*The described robust performance across digestions time is a very positive attribute of this protocol. Other protocols that require MNase digestion are extremely sensitive to its activity and thus very hard to set up. The overlap on size distribution of Figure 1 demonstrate that fragments are of the same size but it is not conclusive for further applications in terms of identity of these fragments. If the size distribution is similar but globally these fragments map to different locations in the genome, the timing on digestion is an extremely important parameter. Although Figure 2—figure supplement 1 attempts to map it, a global mapping and overlap quantification of the data represented for a region in Figure 1 would benefit the understanding of the digestion time importance or lack of it.*

Our intention in showing a doubling digestion time series over >2 orders of magnitude was to vividly illustrate how we get a limit digestion, so that digestion time is *not* critical at all. We now make this point in the text.

*The authors compare Cut&Run with ORGANIC ChIP-seq, a previous generation method from their latest improved method. How does this method compare to their latest method to map TF binding at single base-pair resolution published last year?*

We agree, and now show CTCF MNase X-ChIP-seq data. Results are comparable to results using CUT&RUN.

*Cut&Run relies on MNase digestion. How does Cut&Run perform for pioneer transcription factors that bind to inaccessible, thus resistant regions to light MNAse treatment? Thus, should be discussed and perhaps tested (see below).*

Applying CUT&RUN to a designated pioneer transcription factor is not a suitable way to test whether tethered MNase can “break through” inaccessible chromatin for at least three reasons: 1) By definition the pioneer factors open chromatin when they bind, and so any site that is bound by a pioneer factor when the antibody is added to the nuclei is already opened up. 2) Using ChIP-seq, CTCF binding was shown to inhibit binding of the classical pioneer factor, FoxA1 (PMID: 21151129); this finding implies that CTCF is more penetrant than pioneer factors in breaking through inaccessible chromatin. 3) In the December 1, 2016 issue of Cell, CTCF binding sites were shown to be directly centered over sites of nucleosome occupancy mapped using chemical cleavage, and the authors went on to argue that CTCF is itself a pioneer factor (PMID:27889238).

To address the question of whether CUT&RUN applies to inaccessible regions, we profiled heterochromatin that is marked with H3K27me3. As described in response to reviewer 1, we find that CUT&RUN provides excellent profiles of H3K27me3 that correspond closely to H3K27me3 ChIP but with better dynamic range for the same number of reads (new Figure 6—figure supplement 4), definitively excluding the possibility that such regions are inaccessible to tethered MNase.

*Related, and in light of this comment "When aligned to CTCF motifs found within DNaseI hypersensitive 245 sites, CUT&RUN and X-ChIP-seq CTCF heat maps show strong concordance": How does it compare to whole genome ChIP-seq?*

We now present a direct comparison between CUT&RUN and ENCODE ChIP-seq at all previously identified ENCODE CTCF peaks (new Figure 6—figure supplement 1).

*How does the >150bs run compares to an accessibility experiments?*

This question seems to arise from the misunderstanding concerning targeted versus free MNase referred to above. That is, CUT&RUN does not provide accessibility measurements. Being tethered, MNase specifically releases neighboring nucleosomes. In contrast, MNase-seq gives DNA accessibility, TF occupancy and nucleosome positions in a single experiment. To illustrate this distinction, we now compare CUT&RUN size fractions to MNase-seq data at Abf1 and Reb1 sites (compare Figure 2—figure supplement 1 top and bottom rows of panels). In the bottom panels, we show MNase-seq heat maps over an 8-fold digestion range, where it can be seen that the dynamic range of ≤120-bp and ≥150-bp CUT&RUN signals is much higher than for our previously published MNase-seq signals for both low and high levels of MNase-seq digestion.

*Would a MNase experiment be a better background model for peak calling? If true, then the impact of this method should be reworded.*

No. This question seems to assume that tethered MNase should be cleaving preferentially within accessible regions, whereas in our no-antibody controls, we show that there is no such preference. Again, perhaps what has been confusing is the fact that *free* MNase has a strong AT-rich cleavage bias, but from the results of Figure 4, we see that there is no CUT&RUN digestion of AT-rich yeast centromeres. Therefore, an MNase-seq background would actually *introduce* an AT-bias into CUT&RUN profiles where no such bias exists. Also, as can be seen in Figure 2—figure supplement 1, the dynamic range of MNase-seq is much lower than that of CUT&RUN, and so using MNase-seq as background would have little effect on peak calling.

*CTCF is an atypical TFs in terms of its ability to ChIP. Few novel protocols have been established and tested with CTCF but fail when repeated with other TFs. Thus, I highly recommend to perform and report a Cut&Run experiment with another TF such as homeodomain. This to me is an important and the only wet experiment the authors should perform. The rest of my comments could be addressed by re-analysis of the data.*

To address this concern, we have now added CUT&RUN data for c-Myc and Max transcription factors (Figure 6—figure supplement 3) which have very different properties from those of CTCF, Abf1 or Reb1. Homeodomain proteins are especially sequence non-specific, and so we would not have a gold standard that would allow us to compare performance of CUT&RUN to standard ChIP-seq. Rather, we chose Myc and Max because unlike most other mammalian TFs, Max is the obligate dimerization partner of Myc and therefore, all bona fide Myc sites should have Max occupancy. Based on this rigorous criterion, we find that CUT&RUN performs much better than ENCODE ChIP-seq for Myc and Max in K562 cells (subsection “Low background levels reduce sequencing costs”).

*The claim about recovering 3D interactions is weak and to me there is not enough support to include this claim in the manuscript. First, it needs the MNase background model. The MNase is not covalently bound to CTCF, thus some of the release fragments could be released at similar frequencies in both conditions, while CTCF-interacting elements will be resealed at higher frequencies. Secondly, no technique is perfect, but discussion about transient binding and its implications for the digestion of certain regions should be suggested. Finally, the most important information from 3C techniques is that they identify the binding partners. Cut&Run even in the best case scenario will identify regions that could be in contact with one of the many CTCF binding sites in the genome.*

There is no question about the reality of the two classes of sites, as >90% of all high-scoring CTCF ChIA-PET sites have high-scoring CUT&RUN sites. This correspondence to 3D interaction maps was an unexpected discovery that can explain the nature of the sites that are not seen using Native ChIP, but are also seen using X-ChIP (new Figure 6—figure supplement 1). Regardless of what exact mechanism is responsible, or what potential utility these sites might have in characterizing 3D interactions, the high-resolution profiling of contact sites is a feature inherent to our data that potential users of the method need to be made aware of. Further work beyond the scope of this study will be required to decide whether or not this aspect of CUT&RUN will have practical utility.